# CodeGEMM: A Codebook-Centric Approach to Efficient GEMM in Quantized LLMs

**Gunho Park, Jeongin Bae, Byeongwook Kim, Baeseong Park, Jiwon Ryu,**
**Hoseung Kim, Se Jung Kwon, Dongsoo Lee**

NAVER Cloud
{gunho.park3, dongsoo.lee}@navercorp.com
github.com/naver-aics/codegemm

## Abstract

Weight-only quantization is widely used to mitigate the memory-bound nature of LLM inference. Codebook-based methods extend this trend by achieving strong accuracy in the extremely low-bit regime (e.g., 2-bit). However, current kernels rely on dequantization, which repeatedly fetches centroids and reconstructs weights, incurring substantial latency and cache pressure. We present *CodeGEMM*, a codebook-centric GEMM kernel that replaces dequantization with precomputed inner products between centroids and activations stored in a lightweight *Psumbook*. At inference, code indices directly gather these partial sums, eliminating per-element lookups and reducing the on-chip footprint. The kernel supports the systematic exploration of latency–memory–accuracy trade-offs under a unified implementation. On Llama-3 models, *CodeGEMM* delivers $1.83\times$ (8B) and $8.93\times$ (70B) speedups in the 2-bit configuration compared to state-of-the-art codebook-based quantization at comparable accuracy and further improves computing efficiency and memory subsystem utilization.

## 1 Introduction

Driven by the power-law scaling principle, large language models (LLMs) have grown increasingly larger, delivering remarkable advancements in performance [9, 10]. Notably, open-source models like Llama-3 [4] have reached a scale of 405 billion parameters, demonstrating significant improvements over their predecessors. However, deploying such massive models efficiently in production environments poses substantial challenges. For instance, the 405B model requires approximately 810GB of storage for its parameters alone, exceeding the memory capacity of a single multi-GPU node configured with 8 NVIDIA H100s, which offers a total of 640GB of memory. To address these limitations, extensive research has been dedicated to developing various model compression techniques aimed at reducing model size while minimizing performance degradation [8, 24, 22].

Among compression strategies, quantization has emerged as a particularly effective tool for cutting model size and bandwidth demands with minimal accuracy loss. In large language models (LLMs), challenges with activation quantization—caused by activation outliers—and the significant memory footprint of weight parameters have spurred research into weight-only quantization [6]. These methods focus on quantizing model weights to maximize memory efficiency without compromising performance.

State-of-the-art techniques show that 4-bit weight-only quantization achieves nearly the same performance as unquantized models [13, 12], with notable progress even in 3-bit quantization [21, 1]. However, in extreme low-bit (e.g., 2-bit) settings, uniform quantization suffers from significant performance degradation due to its limited representational capacity. Structured non-uniform methods

39th Conference on Neural Information Processing Systems (NeurIPS 2025).

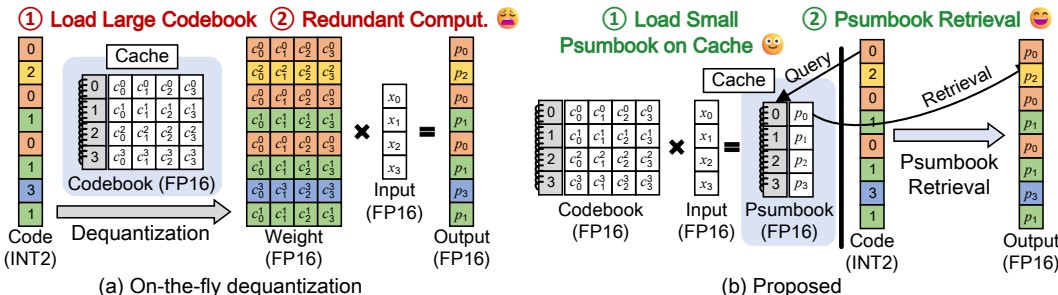

Figure 1: Comparison of matrix multiplication kernels for codebook-based quantized models. The dequantization-based kernel performs on-the-fly dequantization, requiring the entire codebook to be loaded into cache. In contrast, CodeGEMM precomputes partial sums and stores them in a Psumbook, eliminating dequantization overhead and redundant computation.

like Binary-Coded Quantization (BCQ) [30] improve performance over uniform quantization but still fall short of satisfactory results. These challenges emphasize the need for advanced non-uniform quantization techniques capable of maintaining high accuracy in extremely low-bit scenarios.

Codebook-based quantization has emerged as a promising algorithm capable of delivering superior performance in extremely low-bit environments [14, 16, 25, 26, 27]. Unlike traditional quantization methods that represent individual values with fewer bits, codebook-based quantization aims to represent data more efficiently by mapping input vectors to a limited set of centroid vectors. As a result, the original vectors are compressed and stored as codes pointing to the corresponding centroids in the codebook. However, this approach introduces a unique challenge: the efficient management of the codebook. As illustrated in Figure 1(a), unlike uniform quantization, codebook-based quantization relies on the codebook to reconstruct quantized values during computation. Without efficient codebook handling, the overhead associated with dequantization can negate the memory benefits of codebook-based quantization, potentially reducing overall efficiency.

In this paper, we propose *CodeGEMM*, a GEMM method designed to efficiently support codebook-based quantization, enabling practical speedups from extremely low-bit quantization. Unlike existing dequantization-based GEMM kernels, which load the entire codebook into programmable cache memory and rely on dequantization during computation, the proposed kernel precomputes all possible partial sum (Psum) results between the centroid and input data, storing these results as a Psumbook in cache memory (Figure 1(b)). This approach eliminates the necessity for the traditional dequantization step, during which particular centroids are retrieved from the codebook through codes. Instead, the kernel directly retrieves precomputed Psums, reducing redundant computation and significantly decreasing the space complexity required for cache storage. Moreover, the proposed kernel supports a wide range of hyperparameters that define the configuration of codebook-based quantization, facilitating the execution of various quantized models within a unified kernel. This flexibility enables users to explore and evaluate trade-offs among latency, memory usage, and accuracy, providing a versatile and efficient solution for quantized operations.

The contributions of this work are as follows:

1. We introduce *CodeGEMM*, a new approach centered on codebooks to enhance the efficiency of GEMM in codebook-based quantized LLMs. *CodeGEMM* overcomes the drawbacks of existing methods, which require loading the entire codebook into programmable cache memory for dequantization-based operations and suffer from redundant computations.

2. *CodeGEMM* accommodates a broad spectrum of codebook hyperparameters, such as the number of codebooks, vector length, and group size, all within one kernel. This adaptability allows for investigating the trade-offs between latency, memory consumption, and accuracy in codebook-based quantized models, facilitating better optimization for various use cases.

3. On Llama-3.1 models, *CodeGEMM* delivers 1.83× (8B) and 8.93× (70B) speedups in the 2-bit configuration compared to state-of-the-art codebook-based quantization at comparable accuracy, and further improves computing efficiency and memory subsystem utilization.

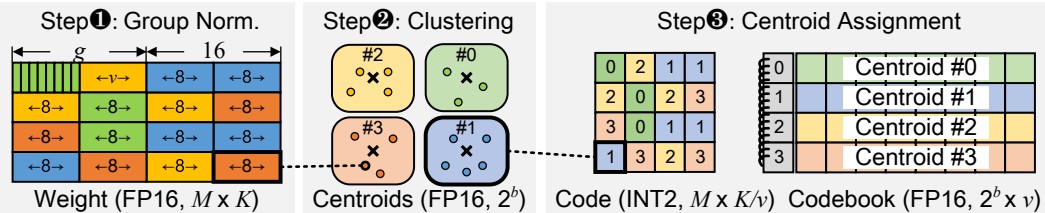

Figure 2: Illustration of quantization process of a $(4 \times 32)$ weight matrix with $b = 2$, $m = 1$, $v = 8$ and $g = 16$.

## 2 Background

### 2.1 Weight-only quantization

As generative language models grow larger and demand greater memory, model quantization has become essential for reducing memory footprints, minimizing model size, and improving inference efficiency. Quantization is commonly applied in two ways: (1) quantizing only the weights or (2) quantizing both weights and activations. However, activation quantization poses significant challenges due to the dynamic range of activation values and extreme outliers, termed *massive activations* [23], which exceed typical values by over $2000\times$.

To address these challenges, many studies have focused on weight-only quantization. GPTQ [6] uses approximate second-order information to compress weights to 4 bits with minimal accuracy loss. AWQ [13] scales salient weights based on activation magnitude to reduce quantization-induced errors. Weight-only quantization enables compact model compression, reducing memory and accelerating inference with specialized kernels [6, 13, 20, 11]. However, at extremely low precision, it still suffers notable accuracy degradation. For instance, QuIP [3] uses random rotation matrices to mitigate this issue, achieving acceptable accuracy but still facing challenges at extremely low-bit levels. To overcome these limitations, non-uniform quantization methods have been introduced [5, 25, 14, 27]. Unlike traditional uniform quantization, non-uniform approaches offer greater flexibility in representing a wider range of values, enhancing accuracy in extremely low-bit quantization scenarios [5].

### 2.2 Codebook-based quantization

Extensive research has been conducted to utilize codebook-based quantization for compressing large language models, utilizing its non-uniform properties and flexibility in representing a wide range of values [5, 16, 25, 26, 27]. In particular, GPTVQ [27] extended the GPTQ framework to incorporate codebook-based quantization, interleaving the quantization of one or more columns with updates to the remaining unquantized weights. Similarly, AQLM [5] introduced an adaptation of multi-codebook quantization for large language models, proposing a method to optimize the codebook using a calibration dataset. QuIP# [25] adopts a smoothening approach similarly to QuIP [3], applying a rotation matrix to transform weights into a space that minimizes worst-case quantization error before mapping them to structured lattice codebooks. Similarly, QTIP [26] builds on this idea by combining rotation-based smoothening with trellis-coded quantization to further enhance performance.

In this work, we build upon the additive codebook quantization strategy adopted in AQLM [5], which represents weights as the sum of multiple centroid vectors. Compared to smoothening-based approaches that rely on matrix transformations such as rotations, additive codebook quantization offers a more intuitive and inference-efficient alternative by avoiding transformation overhead during runtime. This formulation enables fast lookup-based inference and supports flexible trade-offs between accuracy, memory footprint, and latency through careful control of hyperparameters such as the number of codebooks and vector length.

Codebook-based quantization reduces memory footprint by representing the original weights as a set of vectors and approximating each individual vector with a corresponding centroid vector. As illustrated in Figure 2, a weight matrix of size $(M \times K)$ is quantized into codebooks and corresponding codes using several key hyperparameters, including the vector length $(v)$, group size $(g)$, the number of bits per code $(b)$, and the number of codebooks $(m)$. In the first step, the parameter $v$ specifies

the granularity at which the $(M \times K)$ weight matrix is partitioned into vectors. Each vector is then normalized by a scaling factor to facilitate efficient clustering of weight vectors (Step ❶). The normalization group size $g$, which is always greater than or equal to $v$, defines the range over which normalization is performed, ensuring consistent scaling across all vector elements. Specifically, given that the length of each vector is $v$, a total of $g/v$ weight vectors are treated as a single group for normalization purposes.

After normalization, vectors are clustered and mapped to centroid values, which serve as representative values for each cluster (Step ❷). A widely used approach for determining these centroid vectors is the K-means clustering algorithm, which partitions the data into a predefined number of clusters [5, 27, 14]. The algorithm iteratively initializes cluster centroids and assigns each data point to the nearest cluster based on a chosen distance metric, repeating this process until convergence is achieved.

Each centroid vector is assigned a unique index, referred to as a code. The number of clusters is set to $2^b$, where $b$ denotes the number of bits per code, allowing representation of values ranging from 0 to $2^b - 1$. A codebook that maps each code to its corresponding centroid vector is constructed (Step ❸), and original weight values can subsequently be reconstructed by decoding the codes using this codebook. To minimize quantization error, multi-codebook concept has been proposed [5, 2], which performs the quantization process $m$ times, constructing $m$ codebooks and summing their values to represent the original weight value more accurately.

Since applying codebook-based quantization makes $v$ sized full-precision values to be represented by $m$ different codes of $b$ bit precision, the average bit per weight is decided by these hyperparameters. Specifically, the number of bits per weight is calculated by dividing the total quantized bits by the number of weights. The total bit size, and consequently, the average bit per weight $\bar{q}$, can be expressed as follows:

$$
\begin{aligned}
\bar{q} &= \frac{S_{codebook} + S_{code} + S_{norm}}{M \cdot K} \\
&= \frac{16 \cdot m \cdot 2^b \cdot v + b \cdot m \cdot M \cdot \frac{K}{v} + 16 \cdot M \cdot \frac{K}{g}}{M \cdot K},
\end{aligned}
\tag{1}
$$

where the precision of each vector element in the codebook is set to FP16. Equation 1 also reveals that multiple combinations of hyperparameters can achieve the same average bit per weight. For example, as shown in Table 1, various combinations including $(v, m, b, g) = (4, 1, 8, -1)$ and $(v, m, b, g) = (16, 3, 8, 32)$ result in almost the same average bit per weight of 2. Since our goal is to maintain model accuracy under extremely low-bit quantization, achieving an optimal trade-off between bit precision and accuracy is essential. However, the impact of these hyperparameters on model accuracy and inference latency remains largely unexplored, necessitating extensive investigation through comprehensive experimental studies. In this paper, we analyze how different hyperparameter combinations with similar average bits per weight influence the trade-offs between memory footprint and performance, providing valuable insights into optimizing quantized models.

Table 1: Average bits per weight for various quantization configurations. $q_{code}$, $q_{codebook}$, and $q_{norm}$ represent the average bits allocated to codes, codebooks, and group normalization factors, respectively. A value of $g = -1$ indicates row-wise group normalization.

| v | m | b | g | $q_{code}$ | $q_{codebook}$ | $q_{norm}$ | $\bar{q}$ |
|---|---|---|---|---|---|---|---|
| 4 | 1 | 8 | -1 | 2.0 | 0.001 | 0.004 | 2.005 |
| 8 | 2 | 8 | -1 | 2.0 | 0.004 | 0.004 | 2.008 |
| 16 | 4 | 8 | -1 | 2.0 | 0.016 | 0.004 | 2.020 |
| 8 | 1 | 8 | 16 | 1.0 | 0.002 | 1.000 | 2.002 |
| 16 | 3 | 8 | 32 | 1.5 | 0.012 | 0.500 | 2.012 |

## 2.3 Kernels for Quantized LLM

Recent weight-only quantization techniques, aimed at reducing bit-widths to sub-4-bit levels to enhance memory efficiency, are also focused on achieving practical speedups. For instance, GPTQ [6] and AWQ [13] provide INT3-FP and INT4-FP kernels for uniform quantization, respectively, alongside their quantization methodologies. While these kernels achieve reduced latency relative to the

FP16 cuBLAS baseline, they fall short of improving computational efficiency. This limitation arises as the benefits primarily stem from more efficient data movement facilitated by quantization, rather than from the computational performance itself. To address this issue, LUT-GEMM [20] introduces a kernel that reduces computational complexity by utilizing a lookup table, leveraging the BCQ format to represent quantized weights.

Especially in codebook-based quantization, the current kernel [5] relies on a dequantization process. This process reconstructs the original weights by using the codes to retrieve the corresponding centroids from the codebook. These codes serve as indices for querying the codebook, where the corresponding centroids are retrieved and used as dequantized weights. The $(M \times K/v)$ code matrix retrieves centroid vectors of length $v$, reconstructing a dequantized matrix of the same shape as the original matrix $(M \times K)$. Subsequent matrix multiplication with the input matrix is then performed in the same manner as with the original weight matrix. To support efficient dequantization, current kernels load the entire codebook in programmable cache to enable rapid retrieval of centroid vectors. However, this approach leads to significant memory write overhead during the codebook load phase. Furthermore, as the codebook size increases, loading the entire codebook in programmable cache becomes more challenging. For instance, in the AQLM $(1 \times 16)$ kernel the codebook requires $2^{16}$ centroids of length $v = 8$, each represented in FP16. This requires $2^{16} \times 1 \times 8 \times 2$ bytes (=1MB) of shared memory, far exceeding the capacity of both A100 (164KB) and H100 (224KB). As a result, the codebook cannot reside in shared memory and must be repeatedly fetched from DRAM, significantly increasing latency. Furthermore, while this dequantization approach effectively optimizes data movement, it does not reduce computational complexity, which remains identical to that of the original matrix multiplication.

## 3 CodeGEMM: Codebook-based GEMM

**Motivation.** Figure 1 illustrates the matrix multiplication process in codebook-based quantization and highlights two key challenges. First, loading the entire codebook into programmable cache incurs substantial overhead that scales with the size of the codebook and is repeated by each thread block, leading to performance degradation in large-scale deployments. Second, computations are often redundant, as each input vector interacts with a limited set of centroids, resulting in repeated calculations across the output matrix. To address these challenges, we propose a method that minimizes codebook load overhead and avoids redundant computation, enabling more efficient use of limited on-chip cache and improving overall efficiency.

**Methodology.** *CodeGEMM* introduces a codebook-centric approach to efficient matrix multiplication in quantized LLMs. Instead of loading the entire codebook, *CodeGEMM* stores precomputed results of the inner products between the centroid vectors and input activations in the programmable cache. This approach reduces space complexity and eliminates the need for dequantization operations that retrieve centroids for each code from the codebook. Instead, the kernel repeatedly utilizes the precomputed results, significantly reducing the computational complexity.

Figure 3 illustrates the computation process of CodeGEMM, depicting a matrix multiplication operation between a weight tile of dimensions $(t_h \times t_w)$ and an input tile of dimensions $(t_w \times 1)$. Initially, the input tile is partitioned into inputs of dimensions $(t_w/v \times v \times 1)$ to facilitate efficient computation with centroids stored in the codebook (Step ❶). These segmented inputs are represented as $x_k^j = x_{v \times j+k}$, where $j \in [0, 1, \ldots, t_w/v - 1]$ and $k \in [0, 1, \ldots, v - 1]$. The corresponding code matrix is similarly partitioned into code tiles of dimensions $(t_h \times t_w/v)$ to align with the input tiles for effective computation. In our implementation, we set $t_w = 32$ and $t_h = 2048$ to align with hardware-friendly tiling strategies and to maximize reuse of the Psumbook within each thread block. Next, each segmented input computes inner products with the centroids to generate partial sums (Psums), which are then stored as a Psumbook in the programmable cache (Step ❷). Each entry in the Psumbook, $p_i^j$, is calculated as follows:

$$p_i^j = \sum_{k=0}^{v-1} c_k^i \times x_k^j, \quad i \in [0, 1, \ldots, 2^b - 1] \tag{2}$$

By storing the Psumbook in the programmable cache instead of the full codebook, *CodeGEMM* replaces the conventional process of fetching and computing with centroids for each code with a more

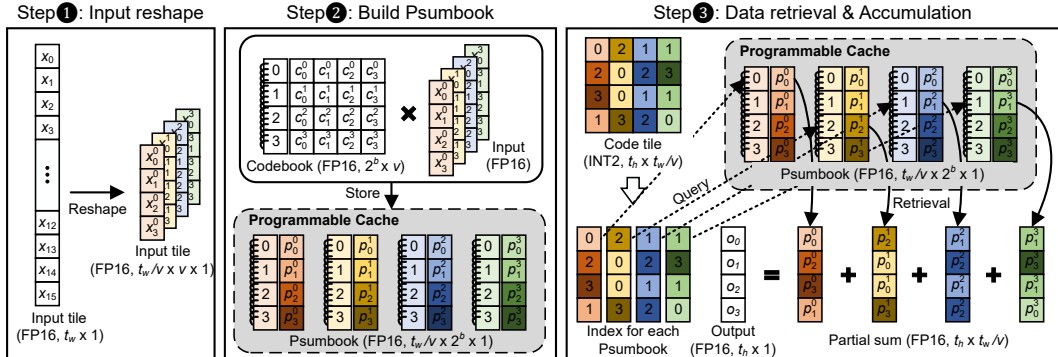

Figure 3: Overview of the *CodeGEMM* kernel operation for codebook-based quantized models. 1) Input data is reshaped into vectors to align with the codebook dimensions. 2) Precomputed inner products between the codebook and input vectors are stored in the Psumbook within the programmable cache, significantly reducing computational overhead. 3) During computation, codes query the corresponding partial sums from the Psumbook, which are then accumulated to generate the output efficiently without requiring on-the-fly dequantization.

efficient retrieval of precomputed Psums using code as a key. This not only reduces computational complexity but also lowers space complexity, as only the scalar inner product results are cached rather than the full centroid vectors. Finally, the partial sums corresponding to each input are retrieved from the Psumbook via code-based indexing and then accumulated to generate the final output activations for the thread block (Step ❸). In contrast to existing kernels that load the entire codebook into the programmable cache, our approach of storing precomputed inner product results allows for significantly more efficient computations for codebook-based quantized LLMs.

**Computational complexity.** In a matrix multiplication operation with a weight matrix of size $(M \times K)$ and an input matrix of size $(K \times N)$, the computational complexity of a standard GEMM kernel is given as $\mathcal{O}(MNK)$. This complexity also applies to dequantization-based kernels, as they improve data movement efficiency through quantization but do not reduce the overall computation cost.

In contrast, *CodeGEMM* reduces the computational workload by precomputing all inner product results between the centroid vectors and input activations, storing them in a Psumbook, and replacing repeated computations with simple retrieval operations. This allows *CodeGEMM* to achieve a lower computational complexity compared to conventional methods. The computational complexity of *CodeGEMM*, assuming $M \gg 2^b$, is expressed as follows:

$$
\begin{aligned}
C = C_{build} + C_{read} &= \mathcal{O}\left(m \cdot 2^b \cdot v \cdot \frac{K}{v} \cdot N\right) + \mathcal{O}\left(m \cdot M \cdot \frac{K}{v} \cdot N\right) \\
&= \mathcal{O}\left(mMNK\left(\frac{2^b}{M} + \frac{1}{v}\right)\right) \approx \mathcal{O}\left(MNK \cdot \frac{m}{v}\right),
\end{aligned}
\tag{3}
$$

where $C_{build}$ and $C_{read}$ represent the computational complexity of building the Psumbook and retrieving values from it, respectively. Consequently, *CodeGEMM* achieves a computational reduction factor of $(m/v)$ compared to conventional kernels, where $m$ is the number of codebooks and $v$ is the vector length—both crucial for optimizing performance. This enables *CodeGEMM* to enhance data movement efficiency through quantization, like traditional dequantization-based kernels, while also reducing computational complexity for higher efficiency.

**Space Complexity.** Dequantization-based GEMM kernels load the entire codebook into the programmable cache to perform computations. In this case, the space complexity is given by $\mathcal{O}(m \cdot 2^b \cdot v)$, which corresponds to the full size of the codebook. In contrast, the space complexity of *CodeGEMM* is $\mathcal{O}(m \cdot 2^b \cdot t_w/v)$, where $t_w$ denotes the width of the weight tile. Since *CodeGEMM* stores precomputed inner product results rather than full centroid vectors, its space complexity is inversely proportional to the vector length $v$, resulting in a smaller cache footprint. This reduction in memory

requirements allows *CodeGEMM* to achieve more efficient cache utilization, making it better suited for accelerators with limited programmable cache sizes.

# 4 Experiments

**Setup.** We evaluate *CodeGEMM* by exploring the trade-offs across key hyperparameters, focusing on three primary metrics relevant to LLM compression: memory footprint, latency, and accuracy. The memory footprint is quantified using the average number of bits per weight $\bar{q}$ and computed according to Equation 1. Latency is measured to compare raw kernel performance for matrix multiplication, assuming the shape of linear layers used in Llama-3 [4], a widely adopted architecture. Specifically, latency is reported as the sum of kernel execution times for all linear layers in a single Transformer decoder block without layer fusion. All latency measurements are performed on an NVIDIA A100 80GB GPU. Throughput (or, equivalently, end-to-end latency) is additionally measured using the Llama implementation provided by the `HuggingFace` [29] library with layer fusion. Although this library is not optimized for high-throughput inference, it remains one of the most widely used frameworks and thus serves as a practical baseline. Accuracy is evaluated using the `lm-eval-harness` [7] benchmark suite across both zero-shot and 5-shot settings on standard tasks.

**Memory Footprint vs. Latency.** Table 2 presents the kernel-level latency of various quantized matrix multiplication methods on 2-bit quantized Llama3 models (8B and 70B). LUTGEMM [20], designed for uniform quantization, achieves strong performance by eliminating redundant computations through efficient use of lookup tables. QuIP# [25] and QTIP [26], which rely on smoothing-based transformations, mitigate the associated overhead through highly optimized fused kernels, demonstrating competitive latency. AQLM [5] with the $2\times8$ configuration improves data movement efficiency via quantization, resulting in moderate latency reduction despite retaining the same computational complexity as the FP16 baseline. However, AQLM-$1\times16$ suffers from significantly higher latency due to inefficient dequantization caused by its large codebook (e.g., $2^{16}$ entries), which exceeds on-chip cache capacity. In contrast, *CodeGEMM* achieves up to $2.18\times$ and $1.64\times$ speed-ups over the FP16 baseline and AQLM, respectively, at the same average bit precision. This improvement is due to both the use of precomputed inner products, which reduce computational complexity, and the efficient utilization of shared memory.

Figure 4(a) shows the relationship between memory footprint and latency of the Llama-3.1-8B model under a single batch operation. As shown in Table 1, codebook-based quantization enables diverse configurations within a given memory footprint by adjusting hyperparameter settings. According to Equation 1, the group size $g$ impacts the memory footprint. Smaller group sizes increase the total memory footprint, leading to higher latency in memory-bound operations. For $g \geq 32$, the fine-grained group normalization incurs minimal latency overhead despite the growing memory footprint. However, as $g$ decreases further, the overhead becomes more pronounced, particularly in per-vector normalization (i.e., $g = v$), where the latency rises sharply. Additionally, this overhead is amplified when the group normalization scale factor constitutes a larger proportion of the total memory footprint, which is more likely when the number of codebooks $m$ is small.

Table 2: Kernel-level latency ($\mu s$) of quantized matrix multiplication in 2-bit quantized Llama3 models (8B and 70B). *CodeGEMM* achieves consistently lower latency than other methods.

|  | cuBLAS (fp16) | LUTGEMM (q2-g128) | QuIP# (e8p) | QTIP (r2) | AQLM (1x16) | AQLM (2x8) | CodeGEMM (m2v8g128) | CodeGEMM (m1v4g128) |
|---|---|---|---|---|---|---|---|---|
| 8B | 332.45 | 160.1 | 162.63 | 189.94 | 645.51 | 250.12 | 172.18 | **152.69** |
| 70B | 1111.36 | 299.9 | 403.59 | 477.04 | 2285.5 | 674.67 | 373.49 | **293.82** |

**Efficiency and Utilization.** We measured DRAM traffic proxies and power efficiency using `nvidia-smi` [18] telemetry. Metrics were sampled at a 100 ms cadence over a 10 s window and averaged across trials. Memory utilization represents the fraction of the sampling interval during which device (global) memory was actively read from or written to [18]. As summarized in Table 3, on a matrix multiplication workload with $M=1$, $N=28672$, and $K=8192$, *CodeGEMM* delivers substantially higher compute efficiency (GFLOPS/W) than dequantization-based kernels. Parenthetical values denote two-sigma error margins over 128 samples. Beyond lower latency, *CodeGEMM* also

shows higher energy efficiency and improved memory-subsystem utilization, indicating reduced and more structured DRAM access relative to dequantization-based approaches.

Table 3: Kernel-level Performance evaluation on a GEMV with $(M, N, K) = (1, 28672, 8192)$. Performance metrics are obtained from `nvidia-smi` telemetry. *CodeGEMM* delivers higher power efficiency and improved memory utilization than dequantization-based codebook kernels.

| Method | TFLOPS | Power (W) | GFLOPS/W | GPU Util (%) | Mem Util (%) |
|---|---|---|---|---|---|
| cuBLAS | 1.58 | 318.55 $(\pm 6.26)$ | 4.95 | 96.87 $(\pm 0.73)$ | 96.94 $(\pm 0.48)$ |
| AQLM-1x16 | 0.75 | 126.54 $(\pm 0.49)$ | 5.93 | 99.00 $(\pm 0.00)$ | 6.00 $(\pm 0.00)$ |
| AQLM-2x8 | 2.59 | 254.20 $(\pm 2.47)$ | 10.18 | 92.84 $(\pm 1.58)$ | 19.96 $(\pm 0.39)$ |
| CodeGEMM-m2v8g128 | 5.43 | 304.69 $(\pm 6.11)$ | 17.83 | 85.32 $(\pm 1.58)$ | 43.75 $(\pm 0.95)$ |
| CodeGEMM-m1v4g128 | 6.12 | 316.38 $(\pm 8.37)$ | **19.36** | 84.47 $(\pm 2.28)$ | 49.80 $(\pm 1.21)$ |

**Memory Footprint vs. Accuracy.** Figure 4(b) shows the relationship between memory footprint and accuracy of quantized models across various hyperparameter configurations. Perplexity, measured using the WikiText-2 dataset, is used to evaluate accuracy. *CodeGEMM* builds on block-wise codebook optimization [5] and incorporates fine-grained group normalization for enhanced performance. As average bits per weight increase, models demonstrate greater representational capacity, resulting in lower perplexity. Fine-grained group normalization reduces quantization error, but it increases memory footprint as the normalization factor grows. Under row-wise group normalization ($g = -1$), increasing the number of codebooks ($m$) while keeping the average bit precision fixed leads to improved accuracy, as the model can better approximate the original weights with additive codebooks. However, as $g$ becomes smaller (i.e., more fine-grained), this accuracy gain from increasing $m$ diminishes, and models tend to exhibit similar performance for a given $\bar{q}$, regardless of the number of codebooks.

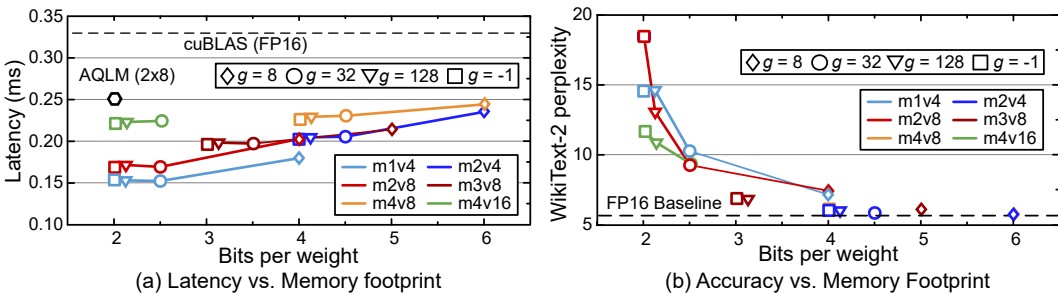

Figure 4: Latency and accuracy trade-offs for the Llama-3.1-8B model under various configurations.

**Throughput vs. Accuracy.** While many studies on quantization have focused on the trade-off between memory footprint and accuracy to achieve better performance, the relationship between throughput (or, equivalently, decode-phase latency) and accuracy has often been overlooked. However, in real-world applications and service-level deployments, latency is often a critical constraint that directly impacts user experience and system efficiency.

Table 4 compares the accuracy of *CodeGEMM* against both uniform and codebook-based quantization methods. We include FlexRound [12] as a representative uniform quantization baseline and AQLM [5] for codebook-based quantization. To further improve accuracy, we also apply PV-Tuning [16], a recently proposed post-quantization calibration method for codebook-based models. Figure 5 reports the resulting accuracy–throughput trade-offs. Throughput for FlexRound is measured using LUT-GEMM [20], a state-of-the-art kernel optimized for uniformly quantized models. While FlexRound demonstrates strong throughput at 2-bit precision, it suffers from the worst accuracy among all methods, highlighting the limitations of uniform quantization in extremely low-bit settings. The recent vector-quantization approach VPTQ [14] achieves an average accuracy of 57.98, slightly higher than *CodeGEMM* without PV-Tuning. However, its kernel is implemented as a straightforward dequantize-then-multiply pipeline without operator fusion, which introduces dequantization overhead and leads to lower throughput than FP16.

AQLM, with its dequantization-based kernel, achieves notably higher accuracy in such settings but exhibits poor throughput due to the overhead of repeatedly loading large codebooks. Specifically, AQLM ($1 \times 16$) achieves the best accuracy under a 2-bit memory budget by using a large codebook ($b = 16$), but this configuration exceeds the capacity of the programmable cache. As a result, it incurs inefficient memory access and dequantization latency, leading to a suboptimal throughput–accuracy trade-off. AQLM ($2\times8$) improves throughput over the FP16 baseline, but only marginally, because it still pays the dequantization cost and maintains similar computational complexity. In contrast, *CodeGEMM* delivers significantly better throughput while maintaining competitive accuracy. Moreover, it shows strong synergy with PV-Tuning, achieving results comparable to or better than AQLM with less computational overhead. Between different *CodeGEMM* configurations, the `m1v4` variant consistently outperforms `m2v8` in both throughput and accuracy, even though both configurations maintain a similar average bit precision. This aligns with the kernel-level latency trend observed in Figure 4(a), while diverging slightly from the perplexity trend in Figure 4(b), suggesting that end-to-end throughput–accuracy behavior can differ from perplexity metrics. Overall, *CodeGEMM* achieves a $1.83\times$ end-to-end speedup over dequantization-based kernels at comparable accuracy (Figure 5(a)).

Table 4: Accuracy of various quantization methods on the Llama-3.1-8B-Instruct model. The Average column represents the mean accuracy across all tasks, including MMLU (5-shot) and and 0-shot tasks such as WinoGrande (WG), HellaSwag (HS), ARC-Easy (ARC-E), and ARC-Challenge (ARC-C).

| Method | $\bar{q}$ | tok/s | MMLU | WG | HS | ARC-E | ARC-C | Avg. |
|---|---|---|---|---|---|---|---|---|
| FP16 | 16.000 | 103.8 | 68.39 | 73.95 | 79.2 | 79.63 | 55.03 | 71.26 |
| FlexRound-q2g128 [12] | 2.125 | 205.3 | 24.27 | 55.16 | 43.78 | 24.75 | 24.57 | 41.65 |
| AQLM-2x8 [5] | 2.005 | 124.5 | 42.29 | 58.25 | 61.40 | 46.25 | 30.89 | 47.82 |
| +PV-Tuning | 2.005 | 124.5 | 55.13 | 69.14 | 72.43 | 71.25 | 45.73 | 62.74 |
| AQLM-1x16 [5] | 2.213 | 49.0 | 58.74 | 68.75 | 70.21 | 73.99 | 46.16 | 63.57 |
| +PV-Tuning | 2.213 | 49.0 | **60.72** | **70.24** | **74.33** | **74.92** | **48.89** | **65.82** |
| **CodeGEMM-m1v4g128** | 2.126 | **228.3** | 45.16 | 58.96 | 63.07 | 63.97 | 38.48 | 53.93 |
| +PV-Tuning | 2.126 | **228.3** | 57.42 | 69.06 | 73.85 | 73.15 | 46.33 | 63.96 |
| **CodeGEMM-m2v8g128** | 2.127 | 214.4 | 42.83 | 60.54 | 62.84 | 60.14 | 37.03 | 52.67 |
| +PV-Tuning | 2.127 | 214.4 | 55.42 | 68.75 | 73.02 | 73.91 | 47.70 | 63.76 |

**Scaling to Larger Models.** To evaluate the scalability of *CodeGEMM* for larger language models, we assess its performance on the Llama-3.1-70B model under various quantization configurations. Table 5 compares *CodeGEMM* with uniform quantization methods such as GPTQ [6] and FlexRound [12], as well as the codebook-based method AQLM. As model size increases from 8B to 70B, latency improvements relative to the AQLM become more pronounced. While the overall performance trends are consistent with those observed in the 8B model, we observe that ($1 \times 16$) suffers from significantly degraded throughput at 70B due to the large codebook size (e.g., $2^{16}$ entries), which exceeds shared memory limits and introduces excessive dequantization overhead. In contrast, *CodeGEMM* achieves a favorable trade-off by leveraging fine-grained group normalization to improve accuracy with minimal increases in memory footprint and latency. The benefit of fine-grained normalization is evident in the widening accuracy gap between *CodeGEMM* and AQLM ($2\times8$), which uses row-wise normalization, at 70B relative to 8B. Consequently, *CodeGEMM* matches the accuracy of AQLM ($1\times16$) while delivering an $8.93\times$ throughput advantage (Figure 5(b)). These results demonstrate that *CodeGEMM* scales effectively to large models while maintaining competitive performance.

## 5 Related Work

**Look-Up Table-Based Computation.** Several prior works have leveraged look-up tables (LUTs) to improve computational efficiency, both at the kernel level [20, 15] and in specialized hardware designs [28, 17, 19]. However, these approaches primarily support uniform or binary-coded quantization formats, and do not extend to codebook-based non-uniform quantization. In contrast, *CodeGEMM* is designed to naturally support codebook-based quantization while also being generalizable to uniform and binary formats through appropriate codebook configurations. Specifically, uniform and binary quantization can be accommodated by defining centroids as $c \in \{0, 1\}^v$ for uniform quantization and

Table 5: Accuracy of various quantization methods on the Llama-3.1-70B model.

| Method | $\bar{q}$ | tok/s | MMLU | WG | HS | ARC-E | ARC-C | Avg. |
|---|---|---|---|---|---|---|---|---|
| FP16 | 16.000 | **OOM** | 78.58 | 79.64 | 85.03 | 86.66 | 64.93 | 78.97 |
| GPTQ-q2g128 | 2.125 | 41.7 | 26.35 | 53.04 | 49.04 | 48.95 | 29.52 | 41.38 |
| FlexRound-q2g128 | 2.125 | 41.7 | 26.70 | 53.59 | 50.67 | 25.13 | 24.83 | 36.58 |
| AQLM-2x8 | 2.002 | 19.0 | 61.45 | 59.59 | 52.83 | 48.82 | 28.67 | 50.27 |
| AQLM-1x16 | 2.055 | 5.5 | **73.07** | **76.16** | **80.83** | **82.20** | **57.17** | **73.89** |
| **CodeGEMM-m1v4g128** | 2.125 | **51.2** | 68.15 | 74.90 | 75.37 | 79.42 | 52.73 | 70.11 |
| **CodeGEMM-m1v4g32** | 2.500 | 49.1 | 71.21 | 76.64 | 79.43 | 82.41 | 56.06 | 73.15 |

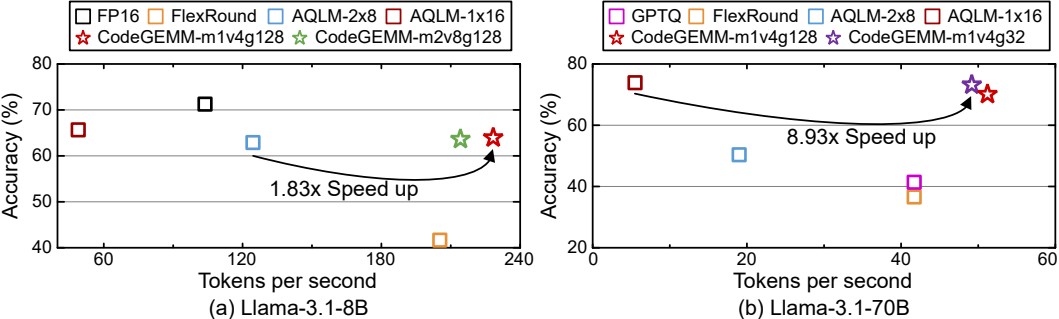

Figure 5: Latency and accuracy trade-offs for the Llama-3.1-8B model under various configurations.

$c \in \{-1, 1\}^v$ for binary quantization. This generality makes *CodeGEMM* particularly suitable for deployment as a fixed-function ASIC kernel, as it enables support for diverse quantization formats under a unified hardware architecture.

**Codebook-based Quantization.** Beyond additive codebooks, recent work has explored highly structured designs that integrate a rotation with the codebook itself. QuIP# [25] and QTIP [26] both pair their lattice- and trellis-coded codebooks with an inference-time *smoothening* rotation, and each work supplies a carefully hand-tuned kernel that fuses the rotation with subsequent look-ups. In contrast, the additive codebooks we target require only a lightweight table lookup and accumulation without rotation, so dequantization remains simple and fast. This design lets *CodeGEMM* deliver competitive 2-bit accuracy while retaining a single, reusable kernel that scales across a broad range of quantization settings.

## 6 Conclusion

**Summary.** This paper presents *CodeGEMM*, an efficient matrix multiplication method for codebook-based quantized models. Instead of loading the full codebook into programmable cache, *CodeGEMM* precomputes and stores results in a Psumbook, reducing both space and computational complexity. Experiments show that *CodeGEMM* outperforms state-of-the-art kernels in latency while maintaining high accuracy in extremely low-bit quantization.

**Limitations.** *CodeGEMM* requires the Psumbook to reside in on-chip shared memory, which constrains the usable codebook size. Very large codebooks (e.g., $b=16$ with $2^{16}$ entries) exceed current GPU cache limits. In practice, we therefore fix the per-codebook width to $b=8$ and recover accuracy via fine-grained group normalization, which adds negligible latency and yields favorable accuracy–efficiency trade-offs under realistic hardware constraints. A second limitation is large-batch throughput: *CodeGEMM* underperforms compared to Tensor Core–based cuBLAS when the batch size is large (e.g., $M>32$). This behavior is common to CUDA Core–based quantized GEMM kernels and reflects the current commercial GPU architecture rather than an inefficiency of the algorithm itself. Overall, while *CodeGEMM* excludes extremely large codebooks and is not optimized for large batches, its hardware–algorithm co-design addresses key inefficiencies of traditional codebook quantization, reducing both computational and space complexity.

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

## A  Appendix

### A.1  Psumbook Build vs. Read Breakdown

We quantify the relative cost of constructing and consuming the *Psumbook* by isolating execution to a single SM and splitting runtime into two phases: building the Psumbook for each tile and reading it during the main accumulate loop. We sweep tile width $t_w$ and batch size $M$ while fixing $(N, K)$ as indicated, and we report the *percentage of execution cycles* devoted to each phase for two kernel variants, `m2v8` and `m1v4`.

Across settings, the build/read split is stable with respect to $M$ at a fixed $t_w$, which indicates that Psumbook construction amortizes across the batch. At a fixed $M$, larger $t_w$ increases the build share for small matrices and decreases it for large matrices. Typical ranges are ∼28–46% build and ∼54–72% read for `m2v8`, and ∼20–42% build and ∼58–80% read for `m1v4`.

Table 6: Cycle share (%) spent on building vs. reading the Psumbook under varying $t_w$ and $M$.

| $M$ | $N$ | $K$ | $t_w$ | Psumbook Phase (%) | Proposed_m2v8 | Proposed_m1v4 |
|---|---|---|---|---|---|---|
| 1 | 4096 | 4096 | 32 | Building / Reading | 30.5 / 69.5 | 20.3 / 79.7 |
| 1 | 4096 | 4096 | 64 | Building / Reading | 33.0 / 67.0 | 28.5 / 71.5 |
| 1 | 4096 | 4096 | 128 | Building / Reading | 31.2 / 68.8 | 30.7 / 69.3 |
| 1 | 8192 | 8192 | 32 | Building / Reading | 45.4 / 54.6 | 41.2 / 58.8 |
| 1 | 8192 | 8192 | 64 | Building / Reading | 45.6 / 54.4 | 39.7 / 60.3 |
| 1 | 8192 | 8192 | 128 | Building / Reading | 28.3 / 71.7 | 29.5 / 70.5 |
| 4 | 4096 | 4096 | 32 | Building / Reading | 30.4 / 69.6 | 20.7 / 79.3 |
| 8 | 4096 | 4096 | 32 | Building / Reading | 30.7 / 69.3 | 20.4 / 79.6 |
| 4 | 8192 | 8192 | 32 | Building / Reading | 45.7 / 54.3 | 41.3 / 58.7 |
| 8 | 8192 | 8192 | 32 | Building / Reading | 46.1 / 53.9 | 41.6 / 58.4 |

### A.2  Tile Size Sensitivity

We revisited our heuristic choices for the tile dimensions and conducted a systematic sweep over $t_w \in \{32, 64, 128\}$ and $t_h \in \{2048, 4096\}$ across representative shapes. We find that $t_h = 2048$ consistently yields the best performance over a broad set of workloads, supporting our original choice. For the horizontal tile, smaller values such as $t_w = 32$ work well for relatively small matrices, whereas $t_w = 64$ tends to perform better on large matrices. We attribute this to coarser tiling reducing kernel-launch overhead and improving partial-sum reduction efficiency at scale. Table 7 summarizes representative

results for $(N, K) \in \{(4096, 4096), (8192, 8192)\}$ at $M{=}1$. Overall, these observations justify our default choice $(t_w, t_h) = (32, 2048)$ for small and medium shapes, with $t_w{=}64$ preferred as matrices grow.

Table 7: Effect of tile dimensions on end-to-end latency ($\mu$s).

| $M$ | $N$ | $K$ | $t_w$ | $t_h$ | Proposed_m2v8 | Proposed_m1v4 |
|---|---|---|---|---|---|---|
| 1 | 4096 | 4096 | 32 | 2048 | 26.57 | 25.07 |
| 1 | 4096 | 4096 | 64 | 2048 | 26.76 | 25.40 |
| 1 | 4096 | 4096 | 128 | 2048 | 29.61 | 26.81 |
| 1 | 4096 | 4096 | 32 | 4096 | 28.95 | 27.60 |
| 1 | 4096 | 4096 | 64 | 4096 | 28.49 | 27.68 |
| 1 | 4096 | 4096 | 128 | 4096 | 37.58 | 32.87 |
| 1 | 8192 | 8192 | 32 | 2048 | 39.04 | 36.02 |
| 1 | 8192 | 8192 | 64 | 2048 | 37.23 | 35.33 |
| 1 | 8192 | 8192 | 128 | 2048 | 40.09 | 38.54 |
| 1 | 8192 | 8192 | 32 | 4096 | 37.78 | 36.17 |
| 1 | 8192 | 8192 | 64 | 4096 | 38.29 | 37.70 |
| 1 | 8192 | 8192 | 128 | 4096 | 45.40 | 42.75 |

## A.3 Effect of Higher Bit Precision

We additionally measured latency for higher average bit precisions using the kernel configuration ($g{=}128$, $b{=}8$, $t_w{=}32$, $t_h{=}2048$). For reference, FP16 cuBLAS latency is included. As expected, increasing the effective bits per weight (via larger numbers of codebooks $m$ or smaller vector length $v$) generally raises latency, and the trend is more pronounced for larger matrices (e.g., $M{=}1$, $N{=}K{=}8192$). Even at higher bit precisions, *CodeGEMM* remains competitive with FP16 baselines while offering a flexible accuracy–efficiency trade-off through the $(m, v)$ configuration.

Table 8: Matrix multiplication latency ($\mu s$) for higher effective bit precisions under ($g{=}128$, $b{=}8$, $t_w{=}32$, $t_h{=}2048$). FP16 cuBLAS is shown for reference.

| $M$ | $N$ | $K$ | $m$ | $v$ | bit | latency |
|---|---|---|---|---|---|---|
| 1 | 4096 | 4096 | N/A | N/A | 16.000 | 28.118 |
| 1 | 4096 | 4096 | 1 | 4 | 2.126 | 25.074 |
| 1 | 4096 | 4096 | 2 | 4 | 4.127 | 27.009 |
| 1 | 4096 | 4096 | 1 | 8 | 1.127 | 24.015 |
| 1 | 4096 | 4096 | 2 | 8 | 2.129 | 26.574 |
| 1 | 4096 | 4096 | 3 | 8 | 3.126 | 27.385 |
| 1 | 4096 | 4096 | 4 | 8 | 4.127 | 29.797 |
| 1 | 8192 | 8192 | N/A | N/A | 16.000 | 95.785 |
| 1 | 8192 | 8192 | 1 | 4 | 2.125 | 36.020 |
| 1 | 8192 | 8192 | 2 | 4 | 4.125 | 49.636 |
| 1 | 8192 | 8192 | 1 | 8 | 1.125 | 31.883 |
| 1 | 8192 | 8192 | 2 | 8 | 2.126 | 39.040 |
| 1 | 8192 | 8192 | 3 | 8 | 3.126 | 47.210 |
| 1 | 8192 | 8192 | 4 | 8 | 4.127 | 58.364 |

## A.4 Batch-Size Sensitivity and Fair cuBLAS Accounting

For fair comparison, we include the dequantization stage when reporting FP16 cuBLAS latency, since dequantization is required to precede GEMM in codebook-based pipelines. Under this accounting, *CodeGEMM* remains competitive even at batch sizes of 8 and 16. In data-center deployments, continuous batching can aggregate requests and increase the effective batch size during decoding; yet, many scenarios still operate with small batches (e.g., on-device inference). The batch-size sensitivity observed in CUDA–core quantized kernels reflects architectural constraints such as occupancy limits and shared-memory bandwidth. This limitation is shared by recent methods (e.g., QuIP# and QTIP) and does not indicate algorithmic inefficiency. Table 9 reports linear latency on Llama-3-8B.

Table 9: Aggregate latency of linear layers ($\mu s$) within a Llama-3-8B decoder block as a function of batch size.

| BS | cuBLAS | Dequant | cuBLAS +Dequant | AQLM (1x16) | AQLM (2x8) | QUIP# (e8p) | QTIP (r2) | Proposed (m2v8) | Proposed (m1v4) |
|---|---|---|---|---|---|---|---|---|---|
| 1 | 332 | 1027 | 1360 | 646 | 250 | 163 | 190 | 172 | 153 |
| 4 | 333 | 1027 | 1361 | 2373 | 794 | 445 | 550 | 491 | 405 |
| 8 | 336 | 1027 | 1364 | 4695 | 1515 | 818 | 1034 | 909 | 744 |
| 16 | 340 | 1027 | 1367 | 9267 | 2959 | 1554 | 1991 | 1748 | 1416 |

## A.5 Additional Benchmarks Across Problem Sizes

We expanded our evaluation to a broad sweep of matrix shapes $(M, N, K)$. Latency is measured end to end on a fixed hardware and software stack, with all kernels compiled under identical toolchains. Table 10 reports the full results. cuBLAS runs on Tensor Cores and tends to maintain relatively stable latency as the batch size $M$ grows, whereas recently proposed quantized kernels—including ours—execute on CUDA cores and often show increased latency with larger $M$. Within this CUDA–core class, *CodeGEMM* consistently performs well on large matrices, where arithmetic intensity and memory reuse are high. In practice, the method remains competitive across diverse $(M, N, K)$ settings and is particularly effective for large-scale matrix multiplications that dominate LLM inference.

Table 10: Kernel latency ($\mu$s) across diverse $(M, N, K)$ configurations.

| M | N | K | cuBLAS | AQLM (1x16) | AQLM (2x8) | **Proposed** (m2v8) | **Proposed** (m1v4) | QUIP# (e8p) | QTIP (r2) |
|---|---|---|---|---|---|---|---|---|---|
| 1 | 2048 | 2048 | 19.82 | 28.84 | 20.55 | 20.75 | 20.66 | 19.47 | 19.44 |
| 4 | 2048 | 2048 | 19.99 | 74.67 | 43.31 | 44.04 | 41.92 | 36.71 | 36.00 |
| 8 | 2048 | 2048 | 19.79 | 135.36 | 73.03 | 75.18 | 69.72 | 59.44 | 57.87 |
| 1 | 8192 | 2048 | 30.57 | 28.84 | 28.83 | 25.94 | 26.70 | 25.52 | 27.08 |
| 4 | 8192 | 2048 | 31.31 | 74.67 | 76.15 | 63.97 | 65.36 | 60.70 | 66.18 |
| 8 | 8192 | 2048 | 31.70 | 135.36 | 138.09 | 115.39 | 116.11 | 107.85 | 118.99 |
| 1 | 2048 | 8192 | 27.52 | 60.47 | 30.93 | 24.28 | 23.81 | 23.44 | 24.90 |
| 4 | 2048 | 8192 | 29.82 | 203.86 | 82.18 | 56.21 | 52.57 | 51.91 | 59.03 |
| 8 | 2048 | 8192 | 28.69 | 396.44 | 149.98 | 98.92 | 90.73 | 89.91 | 103.24 |
| 1 | 4096 | 4096 | 28.00 | 63.13 | 32.28 | 24.76 | 24.97 | 23.96 | 26.74 |
| 4 | 4096 | 4096 | 28.54 | 210.03 | 89.76 | 60.58 | 57.79 | 53.92 | 62.74 |
| 8 | 4096 | 4096 | 28.11 | 396.37 | 165.49 | 108.16 | 103.92 | 93.43 | 110.84 |
| 1 | 14336 | 4096 | 88.67 | 168.12 | 64.76 | 38.85 | 37.51 | 38.91 | 51.30 |
| 4 | 14336 | 4096 | 89.08 | 632.69 | 217.68 | 111.20 | 106.90 | 113.28 | 161.23 |
| 8 | 14336 | 4096 | 89.29 | 1252.55 | 422.89 | 211.37 | 196.68 | 212.55 | 308.37 |
| 1 | 4096 | 14336 | 86.31 | 169.31 | 58.70 | 36.15 | 33.92 | 37.27 | 43.85 |
| 4 | 4096 | 14336 | 86.51 | 635.74 | 193.41 | 103.15 | 92.61 | 106.63 | 133.36 |
| 8 | 4096 | 14336 | 86.49 | 1253.11 | 372.97 | 192.63 | 170.16 | 199.31 | 252.12 |
| 1 | 8192 | 8192 | 96.40 | 188.91 | 62.50 | 37.99 | 35.45 | 38.31 | 49.86 |
| 4 | 8192 | 8192 | 100.41 | 713.24 | 208.11 | 111.00 | 98.66 | 111.08 | 157.26 |
| 8 | 8192 | 8192 | 95.45 | 1408.68 | 402.29 | 207.73 | 184.25 | 208.29 | 299.24 |
| 1 | 28672 | 8192 | 297.74 | 625.53 | 181.54 | 86.48 | 76.71 | 101.98 | 134.03 |
| 4 | 28672 | 8192 | 303.10 | 2462.88 | 684.92 | 305.47 | 264.31 | 366.74 | 492.14 |
| 8 | 28672 | 8192 | 295.11 | 4913.52 | 1355.70 | 597.22 | 514.85 | 718.13 | 970.35 |
| 1 | 8192 | 28672 | 302.42 | 618.61 | 180.38 | 86.20 | 76.50 | 101.13 | 124.90 |
| 4 | 8192 | 28672 | 292.59 | 2437.82 | 679.24 | 305.14 | 263.70 | 361.95 | 455.84 |
| 8 | 8192 | 28672 | 293.69 | 4860.85 | 1344.49 | 596.63 | 515.12 | 710.94 | 897.41 |

