# OpenReview forum: "CodeGEMM: A Codebook-Centric Approach to Efficient GEMM in Quantized LLMs"
_NeurIPS.cc/2025/Conference — NeurIPS 2025 poster_

### Official Review · Reviewer_DEUr · 2025-06-29

**Clarity:** 3
**Significance:** 2
**Originality:** 2
**Rating:** 4
**Confidence:** 4

**Summary:**

This paper introduces CodeGEMM, an optimized matrix multiplication kernel for codebook-based quantized large language models. Traditional codebook-based quantization methods suffer from significant latency due to the need for repeated dequantization and frequent codebook access.

CodeGEMM precompute and storing inner product results between codebook centroids and input data in the programmable cache, eliminating the need for on-the-fly dequantization. The kernel is designed to be flexible across various quantization hyperparameters and model configurations.

**Questions:**

see weaknesses

**Ethical Concerns:**

["NO or VERY MINOR ethics concerns only"]

**Final Justification:**

After reading the author's rebuttal and the comments from other reviewers, I maintain my original score.

**Limitations:**

Provide a small-scale analysis or discussion of possible failure cases where Psumbook overhead could become a new bottleneck (e.g., if codebook structure is highly non-uniform or input batch sizes are extreme).

If possible, add error bars or a variance table to the main throughput/accuracy results.

**Quality:**

3

**Strengths And Weaknesses:**

Strengths
1. The Psumbook-based approach is a practical and innovative solution to a key bottleneck in codebook-based quantization, significantly reducing both space and compute complexity.

2. The kernel is adaptable to a wide range of quantization settings, supporting diverse codebook sizes, vector lengths, and group normalization strategies, making it practical for real deployment.

Weaknesses
1. CodeGEMM is constrained by on-chip cache size, making very large codebooks (e.g., b=16) infeasible. The workaround (smaller codebooks, fine-grained group normalization) is reasonable, but may limit certain applications.

2. The performance improvements are well justified empirically, but the paper could benefit from a more formal treatment of complexity, cache usage, and the limits of the Psumbook design.

3. Figure 1 - Figure 3 appears to be a non-vector image. It would be better to replace it with a vector graphic for improved clarity and scalability.

---

> ### Author Rebuttal · Authors · 2025-07-27
>
> ## Weaknesses
> > CodeGEMM is constrained by on-chip cache size, making very large codebooks (e.g., b=16) infeasible. The workaround (smaller codebooks, fine-grained group normalization) is reasonable, but may limit certain applications.
> - We agree, and this is indeed a known limitation of our current implementation. As noted, very large codebooks such as $b=16$ can offer better accuracy but exceed the capacity of on-chip memory (e.g., GPU shared memory), making them impractical in our current design. However, as shown in Table 3, recent calibration techniques such as PV-Tuning have proven highly effective at mitigating accuracy degradation even under smaller codebooks like $b=8$, which previously showed substantial loss. We are optimistic that as more advanced tuning and quantization algorithms are developed, the remaining accuracy gap will narrow further, without requiring excessively large codebooks.
>
> > The performance improvements are well justified empirically, but the paper could benefit from a more formal treatment of complexity, cache usage, and the limits of the Psumbook design.
> - Thank you for pointing this out. While Section 3 provides a high-level discussion of the complexity, cache behavior, and architectural constraints of the Psumbook design, we agree that a more rigorous and detailed analysis could enhance the clarity and completeness of the paper. Our current version emphasizes conceptual clarity and accessibility to help readers understand the motivation behind CodeGEMM. That said, your suggestion is well taken, and we plan to incorporate a more formal treatment in the final version.
>
> > Figure 1 - Figure 3 appears to be a non-vector image. It would be better to replace it with a vector graphic for improved clarity and scalability.
> - Thank you for pointing this out. We agree that vector graphics would improve readability and scalability. While we are unable to modify the manuscript during the rebuttal phase, we will update these figures with vectorized versions in the camera-ready version if accepted.
>
> > If possible, add error bars or a variance table to the main throughput/accuracy results.
> - We appreciate this suggestion. Although we cannot revise the main figures at this stage, we have annotated all numerical results used during rebuttal with ±2σ error margins to ensure transparency and reliability. We will include proper error bars and/or variance tables in the camera-ready version.
>
> ***We sincerely appreciate the insightful suggestions provided. Throughout the rebuttal period, we have made our best effort to address the reviewers' feedback and strengthen the paper accordingly. That said, we recognize that there may still be areas for improvement. If there are additional suggestions or concerns, we would be more than happy to incorporate them. Thank you again for your thoughtful and constructive comments.***

---

> > ### Comment · Reviewer_DEUr · 2025-08-04
> >
> > After reading the author's rebuttal and the comments from other reviewers, I maintain my original score.

---

> > > ### Author Response · Authors · 2025-08-05
> > >
> > > Thank you for taking the time to review our paper and for considering our rebuttal.
> > > We appreciate your efforts throughout the review process.

---

### Official Review · Reviewer_DeVt · 2025-07-02

**Clarity:** 3
**Significance:** 2
**Originality:** 2
**Rating:** 4
**Confidence:** 4

**Summary:**

CodeGEMM is a GPU GEMM kernel designed for large language models that eliminates the costly dequantization step by pre-computing all inner products between input activations and centroid vectors, storing the results in a small shared-memory “Psumbook.” At inference time, the kernel merely indexes this table with the quantized codes, reducing cache traffic and computational complexity while accommodating a broad range of quantization hyperparameters (bits per code, number of codebooks, vector length, group size) without kernel rewrites. Evaluations on 2-bit Llama-3 (8 B and 70 B) show up to 2.27× kernel-level speedups and 1.8× higher end-to-end throughput over state-of-the-art codebook methods, illustrating that a codebook-centric GEMM can translate the theoretical storage gains of extreme low-bit quantization into real-world LLM inference performance.

**Questions:**

1. Can you provide more benchmark results for the kernel for different problem sizes (M, N, K)?
2. Line 175 says "each input vector interacts with a limited set of centroids". Can you provide any evidence for this assumption?
3. If higher bits per weight were used, what would the speedup be like?

**Ethical Concerns:**

["NO or VERY MINOR ethics concerns only"]

**Final Justification:**

My concerns were addressed by the additional benchmark data. So, I'd like to maintain my score.

**Limitations:**

Yes

**Quality:**

3

**Strengths And Weaknesses:**

Strengths:

- This paper is overall well written and easy to follow. The method and results are clearly presented.
- Motivation and the design of the kernel are visualised with side-by-side diagrams (Figure 1) and a step-wise schematic (Figure 3).
- The authors derive both computational and space complexity, showing CodeGEMM reduces the multiply-accumulate count by a factor ≈ m/v.
- It provides up to 2.27× speed-up compared to other codebook-based quantization methods with comparable accuracy in the 2-bit configuration.


Weaknesses:
- It only provides benchmark results of the kernel under two different model sizes. Since this work is all about kernel optimization, more benchmark results under different problem sizes are supposed to be given.
- Implementation details such as choosing t_w = 32, t_h = 2048 are mentioned but not justified.
- Because the Psumbook must reside in shared memory, very large codebooks (b = 16) are out of scope; the authors themselves list this as a limitation.

---

> ### Author Rebuttal · Authors · 2025-07-27
>
> ## Weaknesses
> > It only provides benchmark results of the kernel under two different model sizes. Since this work is all about kernel optimization, more benchmark results under different problem sizes are supposed to be given.
>
> > Can you provide more benchmark results for the kernel for different problem sizes (M, N, K)?
> - Thank you for the suggestion, and we apologize for not including a more comprehensive set of benchmarks earlier. We have now conducted additional experiments across a wide range of (M, N, K) configurations. (Values in parentheses denote ±2σ error margins of 128 samples.)
> - Unlike cuBLAS, which utilizes Tensor Cores and maintains relatively stable latency across batch sizes, recently proposed quantized kernels, including ours, run on CUDA cores and tend to exhibit increased latency as the batch size (M) grows. Among these state-of-the-art kernels, CodeGEMM consistently demonstrates strong performance on large-scale matrix multiplications, where its compute and memory efficiency are best utilized.
>
> | M | N | K  | cuBLAS | AQLM_m1v8b16  | AQLM_m2v8 | CodeGEMM_m2v8  | CodeGEMM_m1v4| QUIP#_e8p | QTIP|
> |---|------|------|--------|----------|-----------|------------------|------------------|------------------|----------------|
> | 1 | 2048 | 2048 | 19.82 (±0.84)| 28.84 (±4.05)  | 20.55 (±0.52) | 20.75 (±1.43) | 20.66 (±0.90) | 19.47 (±0.85) | ***19.44 (±1.46)***|
> | 4 | 2048 | 2048 | 19.99 (±1.95)| 74.67 (±9.95)  | 43.31 (±0.64) | 44.04 (±0.71) | 41.92 (±1.11) | 36.71 (±1.74) | 36.00 (±0.76)|
> | 8 | 2048 | 2048 | 19.79 (±1.09)| 135.36 (±6.28) | 73.03 (±1.02) | 75.18 (±1.13) | 69.72 (±2.45) | 59.44 (±1.06) | 57.87 (±1.11)|
> | 1 | 8192 | 2048 | 30.57 (±1.36)| 28.84 (±4.05)  | 28.83 (±0.67) | 25.94 (±0.89) | 26.70 (±0.88) | ***25.52 (±0.68)*** | 27.08 (±0.82)|
> | 4 | 8192 | 2048 | 31.31 (±1.33)| 74.67 (±9.95)  | 76.15 (±1.90) | 63.97 (±1.17) | 65.36 (±1.03) | 60.70 (±0.56) | 66.18 (±0.90)|
> | 8 | 8192 | 2048 | 31.70 (±1.54)| 135.36 (±6.28) | 138.09 (±1.00)| 115.39 (±1.34)| 116.11 (±1.61)| 107.85 (±0.89)| 118.99 (±17.80)|
> | 1 | 2048 | 8192 | 27.52 (±1.13)| 60.47 (±18.39) | 30.93 (±0.74) | 24.28 (±0.84) | 23.81 (±1.72) | ***23.44 (±0.81)*** | 24.90 (±1.19)  |
> | 4 | 2048 | 8192 | 29.82 (±1.37)| 203.86 (±10.30)| 82.18 (±1.29) | 56.21 (±1.24) | 52.57 (±1.21) | 51.91 (±1.33) | 59.03 (±0.89)  |
> | 8 | 2048 | 8192 | 28.69 (±1.03)| 396.44 (±13.32)| 149.98 (±1.90)| 98.92 (±0.96) | 90.73 (±1.12) | 89.91 (±1.28) | 103.24 (±0.78) |
> | 1 | 4096 | 4096 | 28.00 (±1.51)| 63.13 (±15.02) | 32.28 (±0.92) | 24.76 (±0.82) | 24.97 (±0.96) | ***23.96 (±0.89)*** | 26.74 (±0.96)  |
> | 4 | 4096  | 4096 | 28.54 (±0.94)| 210.03 (±17.94) | 89.76 (±1.03)  | 60.58 (±1.28) | 57.79 (±1.17) | 53.92 (±1.37) | 62.74 (±0.97) |
> | 8 | 4096  | 4096 | 28.11 (±0.74)| 396.37 (±19.42) | 165.49 (±0.80) | 108.16 (±1.15)| 103.92 (±0.93)| 93.43 (±1.04) | 110.84 (±0.83)|
> | 1 | 14336 | 4096 | 88.67 (±7.59)| 168.12 (±6.50)  | 64.76 (±2.44)  | 38.85 (±1.15) | ***37.51 (±1.24)*** | 38.91 (±0.69) | 51.30 (±1.15) |
> | 4 | 14336 | 4096 | 89.08 (±6.37)| 632.69 (±16.72) | 217.68 (±7.61) | 111.20 (±1.35)| 106.90 (±1.51)| 113.28 (±0.85)| 161.23 (±0.99)|
> | 8 | 14336 | 4096 | 89.29 (±6.19)| 1252.55 (±17.89)| 422.89 (±10.00)| 211.37 (±2.31)| 196.68 (±2.60)| 212.55 (±0.86)| 308.37 (±0.97)|
> | 1 | 4096   | 14336 | 86.31 (±6.76)  | 169.31 (±4.36)    | 58.70 (±0.63) | 36.15 (±1.46) | ***33.92 (±1.12)*** | 37.27 (±1.01) | 43.85 (±0.70)  |
> | 4 | 4096   | 14336 | 86.51 (±6.66)  | 635.74 (±9.95)    | 193.41 (±0.65)| 103.15 (±2.33)| 92.61 (±2.05) | 106.63 (±0.73)| 133.36 (±1.32) |
> | 8 | 4096   | 14336 | 86.49 (±7.32)  | 1253.11 (±11.63)  | 372.97 (±1.77)| 192.63 (±3.34)| 170.16 (±2.65)| 199.31 (±0.83)| 252.12 (±0.94) |
> | 1 | 8192   | 8192  | 96.40 (±7.16)  | 188.91 (±4.24)    | 62.50 (±0.96) | 37.99 (±1.03) | ***35.45 (±0.92)***    | 38.31 (±0.65) | 49.86 (±1.00)  |
> | 4 | 8192   | 8192  | 100.41 (±6.41) | 713.24 (±4.30)    | 208.11 (±0.81)| 111.00 (±1.63)| 98.66 (±1.79) | 111.08 (±0.66)| 157.26 (±0.91) |
> | 8 | 8192   | 8192  | 95.45 (±8.13)  | 1408.68 (±5.48)| 402.29 (±2.42)  | 207.73 (±2.37)| 184.25 (±2.18)| 208.29 (±0.91)| 299.24 (±1.55) |
> | 1 | 28672  | 8192  | 297.74 (±11.93)| 625.53 (±10.30)| 181.54 (±1.83)  | 86.48 (±1.13) | ***76.71 (±0.75)*** | 101.98 (±2.07)| 134.03 (±1.10) |
> | 4 | 28672  | 8192  | 303.10 (±7.94) | 2462.88 (±3.48)| 684.92 (±1.66)  | 305.47 (±1.43)| 264.31 (±1.45)| 366.74 (±3.86)| 492.14 (±1.77) |
> | 8 | 28672  | 8192  | 295.11 (±10.35)| 4913.52 (±6.10)| 1355.70 (±3.20) | 597.22 (±1.71)| 514.85 (±1.72)| 718.13 (±5.03)| 970.35 (±1.71) |
> | 1 | 8192   | 28672 | 302.42 (±6.26) | 618.61 (±2.04) | 180.38 (±1.32)  | 86.20 (±0.84) | ***76.50 (±1.82)*** | 101.13 (±2.12)| 124.90 (±1.03) |
> | 4 | 8192   | 28672 | 292.59 (±6.29) | 2437.82 (±2.30)| 679.24 (±2.01)  | 305.14 (±1.48)| 263.70 (±2.13)| 361.95 (±3.70)| 455.84 (±1.19) |
> | 8 | 8192   | 28672 | 293.69 (±7.43) | 4860.85 (±3.99)| 1344.49 (±1.70) | 596.63 (±2.25)| 515.12 (±3.34)| 710.94 (±4.49)| 897.41 (±1.38) |
>
> > Implementation details such as choosing $t_w = 32, t_h = 2048$ are mentioned but not justified.
> - Thank you for pointing this out. Initially, we chose the tile dimensions $t_w = 32$ and $t_h = 2048$ based on heuristic tuning. However, prompted by your feedback, we conducted a more systematic analysis of these implementation parameters.
> - We found that $t_h = 2048$ consistently provided the best performance across a wide range of workloads, supporting our original choice. Interestingly, for $t_w$, smaller values such as $t_w = 32$ worked well for relatively small matrix sizes, whereas larger values like $t_w = 64$ yielded better performance on large-scale matrix multiplications. We believe this is because larger matrices benefit from coarser tiling, which reduces launch overhead and improves partial sum reduction at scale.
> - We will include a brief discussion of these observations in the implementation section of the final version.
>
> | M | N    | K    | t_w | t_h  | CodeGEMM_m2v8   | CodeGEMM_m1v4   |
> |--|----|----|-----|------|-------------|-------------|
> |1|4096|4096|32 |2048|26.57 (±1.40)|25.07 (±1.28)|
> |1|4096|4096|64 |2048|26.76 (±0.93)|25.40 (±1.08)|
> |1|4096|4096|128|2048|29.61 (±0.83)|26.81 (±0.99)|
> |1|4096|4096|32 |4096|28.95 (±0.88)|27.60 (±0.86)|
> |1|4096|4096|64 |4096|28.49 (±1.29)|27.68 (±1.01)|
> |1|4096|4096|128|4096|37.58 (±1.02)|32.87 (±0.86)|
> |1|8192|8192|32 |2048|39.04 (±1.19)|36.02 (±0.98)|
> |1|8192|8192|64 |2048|37.23 (±0.99)|35.33 (±0.83)|
> |1|8192|8192|128|2048|40.09 (±0.80)|38.54 (±1.43)|
> |1|8192|8192|32 |4096|37.78 (±0.81)|36.17 (±1.18)|
> |1|8192|8192|64 |4096|38.29 (±1.08)|37.70 (±0.67)|
> |1|8192|8192|128|4096|45.40 (±0.70)|42.75 (±1.20)|
>
> > Because the Psumbook must reside in shared memory, very large codebooks (b = 16) are out of scope; the authors themselves list this as a limitation.
> - We agree, and this is indeed a known limitation of our current implementation. As noted, very large codebooks such as b=16 can offer better accuracy but exceed the capacity of on-chip memory (e.g., GPU shared memory), making them impractical in our current design. However, as shown in Table 3, recent calibration techniques such as PV-Tuning have proven highly effective at mitigating accuracy degradation even under smaller codebooks like b=8, which previously showed substantial loss. We are optimistic that as more advanced tuning and quantization algorithms are developed, the remaining accuracy gap will narrow further, without requiring excessively large codebooks.
>
> ## Questions
> > Line 175 says "each input vector interacts with a limited set of centroids". Can you provide any evidence for this assumption?
> - Thank you for pointing this out. The phrasing in Line 175 may have caused confusion. As illustrated in Figure 1 of the paper, each output activation is computed by combining the input activation with all $2^b$ centroids per codebook, meaning there is no restriction on which centroids are involved in the computation.
> - However, when the number of output channels $N$ exceeds the number of centroid combinations $2^b$, which is common in practice, the pigeonhole principle implies that multiple output channels will share the same centroid indices. This leads to reuse opportunities in the partial sum computation. CodeGEMM exploits this by precomputing the Psumbook, which enables efficient reuse of these dot products across multiple output channels.
>
> > If higher bits per weight were used, what would the speedup be like?
> - Thank you for the suggestion. We also conducted latency measurements for higher-bit quantization settings using the kernel configuration ($g = 128, b = 8, t_w = 32, t_h = 2048$). For reference, FP16 cuBLAS latency is also included.
> - As shown in the results, higher average bit precision generally leads to increased latency, especially when the number of codebooks (m) increases. This trend is more pronounced for larger matrix sizes, such as M = 8192. The measurements confirm that even at higher bit precisions, CodeGEMM maintains competitive performance compared to FP16 baselines, while offering a flexible trade-off between accuracy and efficiency depending on the (m, v) configuration.
>
> | M | N | K | m | v | bit | latency (us) |
> |--|----|----|----|----|------|--------------|
> |1|4096|4096|N/A|N/A|16.000|28.118 (±0.993)|
> |1|4096|4096|1|4|2.126|25.074 (±1.281)|
> |1|4096|4096|2|4|4.127|27.009 (±0.685)|
> |1|4096|4096|1|8|1.127|24.015 (±0.870)|
> |1|4096|4096|2|8|2.129|26.574 (±1.400)|
> |1|4096|4096|3|8|3.126|27.385 (±0.701)|
> |1|4096|4096|4|8|4.127|29.797 (±0.545)|
> |1|8192|8192|N/A|N/A|16.000|95.785 (±6.836)|
> |1|8192|8192|1|4|2.125|36.020 (±0.980)|
> |1|8192|8192|2|4|4.125|49.636 (±1.365)|
> |1|8192|8192|1|8|1.125|31.883 (±0.870)|
> |1|8192|8192|2|8|2.126|39.040 (±1.190)|
> |1|8192|8192|3|8|3.126|47.210 (±0.985)|
> |1|8192|8192|4|8|4.127|58.364 (±0.988)|
>
> ***We sincerely appreciate the insightful suggestions provided. If there are additional suggestions or concerns, we would be more than happy to incorporate them.***

---

> > ### Comment · Reviewer_DeVt · 2025-08-06
> >
> > Thanks for providing more data. I am curious if there are any restrictions on the M dimension in the benchmark? All the values of M are very small, and make the problem memory-bound.

---

> > > ### Author Response · Authors · 2025-08-06
> > >
> > > - Thank you for your question and for pointing out the limited range of M values in our benchmarks.
> > > - In contrast to the prefill phase of LLM inference, where large batch sizes (that is, large M) are typical, the **decode phase** inherently operates on small batches.
> > > This is especially true in practical scenarios such as autoregressive generation and asynchronous serving, where tokens are produced one at a time and user requests arrive irregularly.
> > > As a result, optimizing for the small-M regime is crucial for reducing latency in real-world applications.
> > > Our benchmarks focus on this setting because **the latency overhead from full weight dequantization** is substantial at small batch sizes.
> > > - As shown below, the cost of dequantization is more than three times that of cuBLAS GEMM when the batch size is four or fewer, but becomes negligible as the batch size increases.
> > > This supports the commonly adopted strategy, as seen in **vLLM’s AQLM implementation ([code](https://github.com/vllm-project/vllm/blob/a10314c6b35c7bad4320286409d8b5e6d11aa56e/vllm/model_executor/layers/quantization/aqlm.py#L348))**, which applies custom quantized kernels for **small batches (e.g., BS ≤ 6)** and switching to **dequantization + cuBLAS** for larger batches .
> > >
> > > (1) LLaMA 3.1–8B (aggregate latency of linear layers in a transformer decoder)
> > > | BS   | cuBLAS (ms) | Dequant (ms) | Overhead (Dequant/cuBLAS) |
> > > | ---- | ----------- | ------------ | ------------------------- |
> > > | 1    | 0.32967     | 1.02742      | 3.12                      |
> > > | 4    | 0.33505     | 1.02742      | 3.07                      |
> > > | 1024 | 1.86090     | 1.02742      | 0.55                      |
> > > | 8192 | 13.03947    | 1.02742      | 0.08                      |
> > >
> > > (2) LLaMA 3.1–70B
> > > | BS   | cuBLAS (ms) | Dequant (ms) | Overhead (Dequant/cuBLAS) |
> > > | ---- | ----------- | ------------ | ------------------------- |
> > > | 1    | 1.10958     | 3.63055      | 3.27                      |
> > > | 4    | 1.09951     | 3.63055      | 3.30                      |
> > > | 1024 | 6.43143     | 3.63055      | 0.56                      |
> > > | 8192 | 51.37813    | 3.63055      | 0.07                      |
> > >
> > > - The performance advantage of our method in this regime stems from the characteristics of current GPU architectures. Large batches are efficiently processed by dense matrix engines such as **Tensor Cores**, whereas small batches are typically memory-bound and do not fully benefit from such hardware acceleration. Nonetheless, we believe that the CodeGEMM computation pattern can be extended to multi-batch inference when paired with a memory hierarchy and processing unit design tailored to its access characteristics. This presents a promising direction for future accelerator design.
> > >
> > > - We plan to incorporate this clarification into the camera-ready version of the manuscript, if accepted. We sincerely appreciate your time and thoughtful feedback throughout the review process. If there are any further questions or points you would like us to clarify, we would be more than happy to provide additional details.

---

> > > > ### Comment · Reviewer_DeVt · 2025-08-08
> > > >
> > > > In the decoding phase, the batch size is much smaller compared to the prefill phase. But this number can still range from ten to hundreds. From the benchmark results, the codegemm kernel doesn't have performance advantages for batch sizes 4 or 8. How do you justify the results in such cases?

---

> > > > > ### Author Response · Authors · 2025-08-08
> > > > >
> > > > > - Thank you for the thoughtful question.
> > > > > First, we would like to clarify a small misunderstanding in the data.
> > > > > For a fair comparison, the cuBLAS latency should include the additional dequantization overhead.
> > > > > Under this accounting, CodeGEMM shows competitive performance even at batch sizes 8 and 16.
> > > > >
> > > > > ### Llama-3-8B-Total (latency in ms)
> > > > >
> > > > > BS | cuBLAS | Dequant | cuBLAS+Dequant | AQLM_m1v8b16 | AQLM_m2v8 | QUIP#_e8p | QTIP | CodeGEMM_m2v8 | CodeGEMM_m1v4
> > > > > -- | -- | -- | -- | -- | -- | -- | -- | -- | --
> > > > > 1 | 0.33245 | 1.02742 | 1.35987 | 0.64551 | 0.25012 | 0.16263 | 0.18994 | 0.17218 | 0.15269
> > > > > 4 | 0.33343 | 1.02742 | 1.36085 | 2.37348 | 0.79425 | 0.44453 | 0.55039 | 0.49058 | 0.40506
> > > > > 8 | 0.33612 | 1.02742 | 1.36354 | 4.69532 | 1.5153 | 0.81796 | 1.03369 | 0.90903 | 0.74442
> > > > > 16 | 0.33953 | 1.02742 | 1.36695 | 9.26682 | 2.95881 | 1.55398 | 1.99055 | 1.74804 | 1.41579
> > > > >
> > > > >
> > > > >
> > > > > - As you correctly point out, in data center deployments, techniques such as continuous batching are often used to aggregate multiple requests and increase the effective batch size in the decoding phase, sometimes reaching tens or even hundreds. However, there are also important scenarios where throughput gains from batching are not possible. A representative example is on-device inference, where the decoding phase batch size is typically small (e.g., 1–4) [1].
> > > > >
> > > > > - Moreover, the batch-size sensitivity of quantized matmul on GPUs is a well-known challenge shared across many state-of-the-art methods, such as QuIP#[2] and QTIP[3]. This is not due to the ineffectiveness of these algorithms themselves, but rather due to architectural constraints of GPUs (e.g., occupancy limits, limited shared memory bandwidth). We hope that future hardware designs, such as custom ASICs, can alleviate this limitation.
> > > > >
> > > > > - Your question helped us identify a point that could be explained more clearly. If accepted, we will incorporate this clarification into the paper for greater transparency. We welcome any further suggestions you may have.
> > > > >
> > > > > [1] Spector, Benjamin, and Chris Re. "Accelerating llm inference with staged speculative decoding." arXiv preprint arXiv:2308.04623, 2023.
> > > > >
> > > > > [2] Tseng, Albert, et al. "QuIP#: Even Better LLM Quantization with Hadamard Incoherence and Lattice Codebooks." International Conference on Machine Learning, 2024.
> > > > >
> > > > > [3] Tseng, Albert, et al. "Qtip: Quantization with trellises and incoherence processing." Advances in Neural Information Processing Systems, 2024.

---

> > > > > > ### Comment · Reviewer_DeVt · 2025-08-09
> > > > > >
> > > > > > Thanks for the clarification.

---

> > > > > > > ### Author Response · Authors · 2025-08-09
> > > > > > >
> > > > > > > We sincerely appreciate your active participation in the discussion.
> > > > > > > The points you raised prompted us to provide additional explanations and analyses, which we believe have improved the overall clarity and completeness of our submission.

---

### Official Review · Reviewer_wnqh · 2025-07-04

**Clarity:** 2
**Significance:** 1
**Originality:** 2
**Rating:** 3
**Confidence:** 2

**Summary:**

The paper presents CodeGEMM, a new GPU kernel for weight-only, codebook-based quantization in large language models (LLMs). Instead of performing on-the-fly dequantization by loading full centroid vectors into cache, CodeGEMM precomputes all possible inner products between centroids and input subvectors—storing them in a compact Psumbook. During GEMM, it uses each code to index directly into this Psumbook, eliminating repeated dequantization and reducing both computational and space complexity.
The kernel is parameterized by vector length $v$, number of codebooks $m$, bits per code $b$, and tile width $t_w$, enabling exploration of the latency memory accuracy trade-off.

**Questions:**

- Can you provide detailed microbenchmarks separating the time spent building versus reading the Psumbook, especially for varying $t_w$ and batch sizes?
- How does CodeGEMM fare when coupled with codebooks generated by TurboQuant (https://arxiv.org/abs/2504.19874)?
- Have you measured DRAM traffic reductions, and can you provide quantitative comparisons to dequantization-based kernels?
- Do you envision streamed or hierarchical Psumbook designs that allow $b>8$ without cache overflow, and what challenges do these impose?

**Ethical Concerns:**

["NO or VERY MINOR ethics concerns only"]

**Final Justification:**

I thank the authors for their response and have read it carefully. While I acknowledge that some concerns may have been addressed to the satisfaction of other reviewers, my own vote is a borderline rejection.

**Limitations:**

See Weaknesses

**Paper Formatting Concerns:**

- Eq. 1 Caption/Table 1: The notation “g = −1 indicates row-wise group normalization” may confuse readers; consider explicit table footnote formatting.
- In Table 3’s caption: “including MMLU (5-shot) and and 0-shot tasks such as…” — the repeated “and” should be fixed

**Quality:**

2

**Strengths And Weaknesses:**

**Strengths:**
- Precomputing and caching inner products removes the need to fetch full centroids for each code, directly addressing the shared-memory capacity bottleneck in existing kernels
- Kernel-level benchmarks demonstrate up to 2.18× improvement over FP16 cuBLAS and 1.64× over AQLM on A100 GPUs, and throughput evaluations confirm real-world speedups in the HuggingFace Llama pipeline

**Weaknesses:**
- The reliance on fitting the Psumbook in shared memory forces $b\le8$ and precludes exploration of larger codebook widths (e.g., $b=16$), limiting accuracy potential in extreme low-bit regimes .
- The focus is on latency and perplexity; energy consumption and DRAM bandwidth savings which are critical in production are unreported.
- It remains unclear how much time is spent constructing the Psumbook versus retrieving entries during GEMM. A breakdown would clarify whether build overheads offset retrieval gains at different batch sizes or tile shapes.
- While NeurIPS checklist items are cited, the paper omits error bars (claimed negligible variance) and details on code release versioning, which may hinder exact reproduction .

---

> ### Author Rebuttal · Authors · 2025-07-28
>
> ## Weaknesses
> > The reliance on fitting the Psumbook in shared memory forces $b\le8$ and precludes exploration of larger codebook widths (e.g., $b=16$), limiting accuracy potential in extreme low-bit regimes .
> - We agree, and this is indeed a clear limitation of our current work. We believe that model compression serves different goals depending on the deployment scenario. If the primary objective is to minimize memory footprint with minimal accuracy degradation, then using larger codebooks such as b = 16 is certainly effective.
> - However, when practical speedup is also a goal, algorithm-hardware co-design becomes essential. To support this, we deliberately introduced constraints during the design of our compression algorithm to ensure compatibility with on-chip memory and efficient kernel execution.
> - As shown in Table 3, recent calibration methods such as PV-Tuning offer promising ways to recover much of the accuracy lost under aggressive quantization. Notably, PV-Tuning was particularly effective in the b = 8 regime, where degradation was previously much higher compared to b = 16. We expect that as more advanced tuning and compression algorithms emerge, the remaining accuracy gap will continue to shrink even under shared memory constraints.
>
> > While NeurIPS checklist items are cited, the paper omits error bars (claimed negligible variance) and details on code release versioning, which may hinder exact reproduction .
> - We apologize for the omission. To support reproducibility, we provide the exact version information of the software environment used in our experiments: python==3.12.3, CUDA==12.6, torch==2.6.0, transformers==4.46.3, lm-eval==0.4.5
> - We also fully agree that including error bars improves transparency. For this rebuttal, we have added 2-sigma error margins over 128 measurement samples to all reported numbers in our responses to the reviewers. While we are unable to modify the manuscript at this stage, we will include error bars in all relevant results in the camera-ready version.
>
> ## Questions
>
> > The focus is on latency and perplexity; energy consumption and DRAM bandwidth savings which are critical in production are unreported.
>
> > Have you measured DRAM traffic reductions, and can you provide quantitative comparisons to dequantization-based kernels?
> - Thank you for the suggestion. We agree on the importance of hardware utilization and conducted a set of experiments to compare the efficiency of different kernels.
> - To evaluate DRAM traffic and power efficiency, we used nvidia-smi queries [1] sampled every 100 milliseconds for 10 seconds, then averaged the results. The results show that CodeGEMM achieves significantly better compute efficiency (FLOPS per watt) compared to dequantization-based kernels.
> - The table below summarizes the results for various methods on a matrix multiplication workload with M = 1, N = 28672, and K = 8192. All values in parentheses indicate 2-sigma error margins over 128 measurement samples.
>
> | Method         | Latency (us)     | TFLOPS | Power (W)         | GFLOPS/W | GPU Util (%)      | Mem Util (%)      |
> |----------------|------------------|--------|--------------------|----------|--------------------|--------------------|
> | cuBLAS         | 297.74 (±11.93)  | 1.58   | 318.55 (±6.26)     | 4.95     | 96.87 (±0.73)      | 96.94 (±0.48)      |
> | aqlm_1x16       | 625.53 (±10.30)  | 0.75   | 126.54 (±0.49)     | 5.93     | 99.00 (±0.00)      | 6.00 (±0.00)       |
> | aqlm_2x8        | 181.54 (±1.83)   | 2.59   | 254.20 (±2.47)     | 10.18    | 92.84 (±1.58)      | 19.96 (±0.39)      |
> | codegemm_m2v8   | 86.48 (±1.13)    | 5.43   | 304.69 (±6.11)     | 17.83    | 85.32 (±1.58)      | 43.76 (±0.95)      |
> | codegemm_m1v4   | 76.71 (±0.75)    | 6.12   | 316.38 (±8.37)     | **19.36**    | 84.47 (±2.28)      | 49.80 (±1.21)      |
>
> - These results highlight that CodeGEMM not only achieves lower latency but also delivers higher energy efficiency and better memory subsystem utilization, suggesting reduced and more structured DRAM access compared to dequantization-based approaches. We will include this analysis in the final version.
>
> [1] https://docs.nvidia.com/deploy/nvidia-smi/index.html
>
> > Can you provide detailed microbenchmarks separating the time spent building versus reading the Psumbook, especially for varying $t_w$ and batch sizes?
> - Thank you for the insightful suggestion. While it is challenging to measure precise cycle-level timing for each kernel phase due to concurrent execution across multiple SMs on the GPU, we performed controlled measurements by isolating execution to a single SM in order to estimate the relative cost of building and reading the Psumbook.
> - The table below reports the percentage of execution cycles spent on each phase for various tile widths $t_w$ and batch sizes M. For a fixed $t_w$, increasing M shows that the ratio between building and reading remains largely stable, indicating that the Psumbook construction cost is effectively amortized across the batch. For a fixed M, increasing $t_w$ tends to increase the proportion of time spent on building in small matrices, but decrease it in large matrices.
>
> | M | N    | K    | $t_w$ | Psumbook Phase (%)       | m2v8         | m1v4         |
> |---|------|------|------|--------------------------|--------------|--------------|
> | 1 | 4096 | 4096 | 32   | Building / Reading       | 30.5 / 69.5  | 20.3 / 79.7  |
> | 1 | 4096 | 4096 | 64   |                          | 33.0 / 67.0  | 28.5 / 71.5  |
> | 1 | 4096 | 4096 | 128  |                          | 31.2 / 68.8  | 30.7 / 69.3  |
> | 1 | 8192 | 8192 | 32   |                          | 45.4 / 54.6  | 41.2 / 58.8  |
> | 1 | 8192 | 8192 | 64   |                          | 45.6 / 54.4  | 39.7 / 60.3  |
> | 1 | 8192 | 8192 | 128  |                          | 28.3 / 71.7  | 29.5 / 70.5  |
> | 1 | 4096 | 4096 | 32   |                          | 30.5 / 69.5  | 20.3 / 79.7  |
> | 4 | 4096 | 4096 | 32   |                          | 30.4 / 69.6  | 20.7 / 79.3  |
> | 8 | 4096 | 4096 | 32   |                          | 30.7 / 69.3  | 20.4 / 79.6  |
> | 1 | 8192 | 8192 | 32   |                          | 45.4 / 54.6  | 41.2 / 58.8  |
> | 4 | 8192 | 8192 | 32   |                          | 45.7 / 54.3  | 41.3 / 58.7  |
> | 8 | 8192 | 8192 | 32   |                          | 46.1 / 53.9  | 41.6 / 58.4  |
>
> > How does CodeGEMM fare when coupled with codebooks generated by TurboQuant (https://arxiv.org/abs/2504.19874)?
> - Thank you for the valuable suggestions. TurboQuant is an online vector quantization method primarily designed for KV‑cache compression. This aligns well with CodeGEMM, which dynamically operates on precomputed codebooks and codes, replacing on-the-fly dequantization with psum lookups. CodeGEMM is agnostic to how the codebooks are trained; it only requires the centroids, codes, and the associated hyperparameters (m, b, v) to construct a Psumbook for the current input tile. As such, codebooks generated by TurboQuant can be directly used with CodeGEMM, provided that the per-codebook bitwidth and vectorization choices fit within on-chip memory constraints.
> - While TurboQuant is designed for online KV‑cache quantization, CodeGEMM naturally extends to this use case. In attention matmuls with quantized activations, the Psumbook is constructed between the current query tile and the key or value centroids, followed by a lookup using the stored codes. This maintains the same memory and compute characteristics, although the Psumbook must be rebuilt for each new query tile.
>
>
> > Do you envision streamed or hierarchical Psumbook designs that allow $b>8$ without cache overflow, and what challenges do these impose?
> - We agree that fitting the entire Psumbook in shared memory limits the maximum codebook size $b$. We are exploring two complementary extensions: streamed or hierarchical Psumbook designs. First, a streamed Psumbook that partitions the table along the codebook dimension and prefetches chunks from global memory with double buffering, overlapping data transfer and lookup. This avoids cache overflow but introduces additional bandwidth demand and synchronization overhead. Second, a hierarchical Psumbook that decomposes a wide codebook into coarse and residual sub-codebooks. Partial sums are built separately for each level and then aggregated, which reduces the $2^b$ memory blow-up at the cost of extra lookups and combination logic. The main challenges include hiding global memory latency, preventing bank conflicts under larger tiles, and managing register and shared memory usage to sustain occupancy. Additionally, if the computational overhead of Psumbook building grows beyond a certain point, the net performance gain of CodeGEMM may diminish.
>
> ***We sincerely appreciate the insightful suggestions provided. Throughout the rebuttal period, we have made our best effort to address the reviewers' feedback and strengthen the paper accordingly. That said, we recognize that there may still be areas for improvement. If there are additional suggestions or concerns, we would be more than happy to incorporate them. Thank you again for your thoughtful and constructive comments.***

---

### Official Review · Reviewer_2GP4 · 2025-07-07

**Clarity:** 3
**Significance:** 2
**Originality:** 3
**Rating:** 5
**Confidence:** 5

**Summary:**

The paper proposes a codebook-based quantization method to represent weights using 2-bit configurations. Unlike previous approaches that stored the entire codebook in memory for dequantization, the proposed method stores the product-sum results, eliminating the need for dequantization. Hyperparameters for codebook selection such as group sizes, vector length, and number of codebooks are explored to find the best trade-off between performance accuracy and hardware complexity. The new approach achieves a 2.29× speed improvement compared to previous codebook-based quantization methods such as AQLM (2×8).

**Questions:**

Is there any limitation on the size of the product-sum values stored in memory for large language models such as LLaMA 3.1 with 405B parameters?

**Ethical Concerns:**

["NO or VERY MINOR ethics concerns only"]

**Final Justification:**

The author provides new experiments to compare with the previous work and also addressed my question regarding the limitations of the new approach for large language models such LLaMA 3. Therefore, I raised my score.

**Limitations:**

yes

**Paper Formatting Concerns:**

There are no concerns regarding the paper format.

**Quality:**

2

**Strengths And Weaknesses:**

Strengths:

1- The new approach is explored across various hardware constraints such as memory footprint, latency, and throughput, and it demonstrates the benefits of product-sum codebook-based quantization compared to previous approaches.

2- The paper is well-written, and the experiments are elaborated in detail.

Weaknesses:

1- The accuracy degradation of the codebook-based approach is high compared to the FP16 approach in Table 3 (approximately 7%). The author is advised to elaborate on this accuracy degradation.

2- It is suggested to include accuracy vs. latency and accuracy vs. throughput plots corresponding to Table 3 and Table 4.

3- It is recommended that the comparison with these previous approaches [1, 2] be demonstrated in the paper.

[1] Liu, Yifei, et al. "Vptq: Extreme low-bit vector post-training quantization for large language models." arXiv preprint arXiv:2409.17066 (2024).

[2] Liu, Zechun, et al. "ParetoQ: Scaling Laws in Extremely Low-bit LLM Quantization." arXiv preprint arXiv:2502.02631 (2025).

---

> ### Author Rebuttal · Authors · 2025-07-27
>
> ## Weaknesses
>
> > 1- The accuracy degradation of the codebook-based approach is high compared to the FP16 approach in Table 3 (approximately 7%). The author is advised to elaborate on this accuracy degradation.
>
> - Thank you for the insightful feedback. We agree with your observation. In the context of LLM post-training quantization, 2-bit quantization still leads to notable accuracy degradation, especially under extreme compression. While advanced techniques such as PV-Tuning help mitigate this issue to some extent, a performance gap remains. We believe that with sufficient resources, quantization-aware training could be a promising direction to further narrow this gap.
> - Our primary goal with CodeGEMM is to serve as a practical kernel-level bridge between model compression and real-world speedup. We are optimistic that as better quantization algorithms continue to emerge, CodeGEMM will enable them to realize their full latency-throughput benefits on modern hardware.
>
> > 2- It is suggested to include accuracy vs. latency and accuracy vs. throughput plots corresponding to Table 3 and Table 4.
> - We agree that figures are better suited than tables for illustrating trade-offs. Although we are unable to modify the manuscript during the rebuttal phase, we will include these plots in the camera-ready version, if accepted. We believe the visual representation will clearly highlight the accuracy-efficiency Pareto front achieved by CodeGEMM under different quantization settings.
>
> > 3- It is recommended that the comparison with these previous approaches [1, 2] be demonstrated in the paper.
> - Thank you for the helpful suggestion. While the reviewer recommended evaluations on both ParetoQ and VPTQ, we were only able to conduct experiments with VPTQ due to the lack of publicly available LLaMA 3 checkpoints for ParetoQ. VPTQ, which is also a vector-quantization-based method, shows comparable accuracy. However, similar to AQLM, it relies on dequantization-based kernels and therefore exhibits lower throughput than the FP16 baseline. We appreciate the suggestion, which helped enrich our study, and we plan to include this result in the camera-ready version if accepted.
>
> | Method                      | 𝑞̄     | tok/s | MMLU  | WG    | HS    | ARC-E | ARC-C | Avg.  |
> |----------------------------|--------|-------|-------|-------|-------|--------|--------|--------|
> | FP16                       | 16.000 | 34.3  | 68.39 | 73.95 | 79.2  | 79.63 | 55.03 | 71.26 |
> | FlexRound-q2-g128 [12]     | 2.125  | 34.4  | 24.27 | 55.16 | 43.78 | 24.75 | 24.57 | 41.65 |
> | AQLM-2x8 [5]               | 2.005  | 20.6  | 42.29 | 58.25 | 61.40 | 46.25 | 30.89 | 47.82 |
> | +PV-Tuning                 | 2.005  | 20.6  | 55.13 | 69.14 | 72.43 | 71.25 | 45.73 | 62.74 |
> | AQLM-1x16 [5]              | 2.213  | 20.8  | 58.74 | 68.75 | 70.21 | 73.99 | 46.16 | 63.57 |
> | +PV-Tuning                 | 2.213  | 20.8  | **60.72** | **70.24** | **74.33** | **74.92** | **48.89** | **65.82** |
> | CodeGEMM-m1-v4-g128    | 2.126  | 36.8  | 45.16 | 58.96 | 63.07 | 63.97 | 38.48 | 53.93 |
> | +PV-Tuning                 | 2.126  | **36.8**  | 57.42 | 69.06 | 73.85 | 73.15 | 46.33 | 63.96 |
> | CodeGEMM-m2-v8-g128    | 2.127  | 36.2  | 41.53 | 55.72 | 62.08 | 65.07 | 39.51 | 52.78 |
> | +PV-Tuning                 | 2.127  | 36.2  | 56.64 | 69.06 | 72.58 | 73.06 | 46.76 | 63.62 |
> | **VPTQ**                       | 2.300  | 31.7 | 44.87 | 69.06 | 70.83 | 64.94 | 40.19 | 57.98 |
>
>
> ## Questions
> > Is there any limitation on the size of the product-sum values stored in memory for large language models such as LLaMA 3.1 with 405B parameters?
> - Our approach is independent of the total model size and scales well to extremely large models such as LLaMA 3.1–405B. The only memory constraint lies in the size of on-chip memory used to store the Psumbook. The size of Psumbook depends only on codebook parameters (e.g., number of codebooks and vector length), not on the number of parameters. Importantly, the performance of CodeGEMM improves with larger problem sizes. As shown in the table below, CodeGEMM achieves greater speedup over cuBLAS as the matrix dimensions grow. Each matrix shape corresponds to configurations used in LLaMA 3 models of varying sizes, including 1B, 8B, 70B, and 405B. (Values in parentheses indicate ±2σ error over 128 measurement samples.)
>
> | M | N     | K     | cuBLAS ($\mu s$)     | CodeGEMM_m2v8 ($\mu s$) | Speed up | CodeGEMM_m1v4 ($\mu s$) | Speed up |
> |---|--------|--------|------------------|----------------------|-----------|----------------------|-----------|
> | 1 | 2048  | 2048  | 20.1 (±0.9)       | 20.8 (±1.4)           | ×1.0      | 20.7 (±0.9)           | ×1.0      |
> | 1 | 8192  | 2048  | 31.2 (±1.3)       | 25.9 (±0.9)           | ×1.2      | 26.7 (±0.9)           | ×1.2      |
> | 1 | 2048  | 8192  | 27.6 (±1.2)       | 24.3 (±0.8)           | ×1.1      | 23.8 (±1.7)           | ×1.2      |
> | | | | | | | | |
> | 1 | 4096  | 4096  | 27.8 (±1.5)       | 24.8 (±0.8)           | ×1.1      | 25.0 (±1.0)           | ×1.1      |
> | 1 | 14336 | 4096  | 88.8 (±6.7)       | 38.8 (±1.1)           | ×2.3      | 37.5 (±1.2)           | ×2.4      |
> | 1 | 4096  | 14336 | 86.2 (±6.9)       | 36.2 (±1.5)           | ×2.4      | 33.9 (±1.1)           | ×2.5      |
> | | | | | | | | |
> | 1 | 8192  | 8192  | 96.4 (±7.2)       | 38.0 (±1.0)           | ×2.5      | 35.4 (±1.1)           | ×2.7      |
> | 1 | 28672 | 8192  | 297.9 (±11.5)     | 86.5 (±1.1)           | ×3.4      | 76.7 (±0.8)           | ×3.9      |
> | 1 | 8192  | 28672 | 302.0 (±6.3)      | 86.2 (±0.8)           | ×3.5      | 76.5 (±1.8)           | ×3.9      |
> | | | | | | | | |
> | ***1*** | ***16384*** | ***16384*** | 340.6 (±8.4)      | 97.2 (±1.3)           | ×3.5      | 97.0 (±2.4)           | ×3.5      |
> | ***1*** | ***53248*** | ***16384*** | 1023.5 (±10.3)    | 263.9 (±1.9)          | ×3.9      | 263.7 (±2.4)          | ×3.9      |
> | ***1*** | ***16384*** | ***53248*** | 1060.0 (±10.8)    | 263.9 (±1.3)          | ×4.0      | 263.8 (±1.2)          | ×4.0      |
>
>
> ***We sincerely appreciate the insightful suggestions provided. Throughout the rebuttal period, we have made our best effort to address the reviewers' feedback and strengthen the paper accordingly. That said, we recognize that there may still be areas for improvement. If there are additional suggestions or concerns, we would be more than happy to incorporate them. Thank you again for your thoughtful and constructive comments.***

---

> > ### Author Response · Authors · 2025-08-06
> >
> > Dear Reviewers,
> >
> > Thank you again for your thoughtful time and feedback on our submission.
> >
> > We have submitted detailed responses to your comments and would greatly appreciate it if you could take a moment to review them when convenient. If there are any questions or further points needing clarification, we are more than happy to engage in discussion.
> >
> > We look forward to your continued guidance.
> >
> > Best regards,
> >
> > The Authors

---

### Note · Authors · 2025-08-11

We sincerely thank the reviewers and the AC for their time and engagement throughout the review process.

In this work, we proposed ***CodeGEMM***, a practical and innovative kernel-level solution to a key bottleneck in codebook-based quantization. By adopting a Psumbook-based approach, CodeGEMM significantly reduces both computational and space complexity, while being explored across diverse hardware constraints such as memory footprint, latency, and throughput.

As noted both in the paper and by several reviewers, CodeGEMM is constrained by on-chip cache size, making very large codebooks (e.g., b = 16) infeasible. However, such large codebooks incur substantial latency overhead. Through hardware–algorithm co-design, we proposed efficient workarounds, such as smaller codebooks and fine-grained group normalization, that achieve favorable accuracy–efficiency trade-offs in practice.

We acknowledge that CodeGEMM underperforms Tensor Core–based cuBLAS in large-batch operations (e.g., batch size > 32). However, this is a shared limitation among CUDA Core–based quantized matrix multiplication kernels and reflects the structural constraints of current commercial GPU architectures, rather than an inefficiency in the algorithm itself.

Overall, CodeGEMM addresses the inefficiencies of traditional codebook-based quantization through co-design, providing a kernel that achieves meaningful reductions in both computational complexity and memory usage.

We once again thank the reviewers for their constructive feedback, which has helped us further clarify and strengthen our work.

---

### Decision · Program_Chairs · 2025-09-17

**Decision:**

Accept (poster)

**Comment:**

The work introduces a matrix multiplication kernel for codebook-based quantized models. Unlike traditional methods that incur latency by repeatedly loading the codebook for dequantization, the proposed approach CodeGEMM eliminates the dequantization step by precomputing inner products into a Psumbook. It supports flexible hyperparameters to balance latency, memory, and accuracy. Experiments show that CodeGEMM achieves a speed-up over state-of-the-art codebook-based quantization methods while maintaining comparable accuracy in 2-bit settings.

This is an interesting algorithm-hardware co-design work. We decided to accept this paper and also suggests the following changes:
- Explicitly point out the limitation of this approach in the final version, e.g., the coodbook size cannot be too large; why CodeGEMM underperforms Tensor Core–based cuBLAS in large-batch operations.
- Narrow down the application area of the proposed approach and retune the claim in the abstract
- Include the additional experimental comparison in the final version.